# DIFFERENTIABLE SOLVER SEARCH FOR FAST DIFFUSION SAMPLING

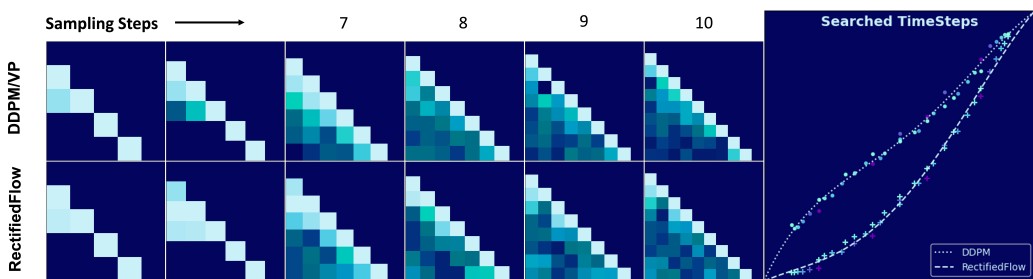

Figure 1: **Visualization of searched Solver Parameters of DDPM/VP and Rectified Flow.** We limited the order of solver coefficients of the last two steps for 5/6 NFE. The left images show the absolute value of searched coefficients $\{c_i^j\}$. The right image shows the searched timesteps of different NFE and fitted curves.

## ABSTRACT

Diffusion models have demonstrated remarkable generation quality but at the cost of numerous function evaluations. Recently, advanced ODE-based solvers have been developed to mitigate the substantial computational demands of reverse-diffusion solving under limited sampling steps. However, these solvers, heavily inspired by Adams-like multistep methods, rely solely on t-related Lagrange interpolation. We show that t-related Lagrange interpolation is suboptimal for diffusion model and reveal a compact search space comprised of time steps and solver coefficients. Building on our analysis, we propose a novel differentiable solver search algorithm to identify more optimal solver. Equipped with the searched solver, rectified-flow models, e.g., SiT-XL/2 and FlowDCN-XL/2, achieve FID scores of 2.40 and 2.35, respectively, on ImageNet-256 × 256 with only 10 steps. Meanwhile, DDPM model, DiT-XL/2, reaches a FID score of 2.33 with only 10 steps. Notably, our searched solver outperforms traditional solvers by a significant margin. Moreover, our searched solver demonstrates generality across various model architectures, resolutions, and model sizes.

## 1 INTRODUCTION

Image generation is a fundamental task in computer vision research, which aims at capturing the inherent data distribution of original image datasets and generating high-quality synthetic images through distribution sampling. Diffusion models Ho et al. (2020); Song et al. (2020b); Karras et al. (2022); Liu et al. (2022); Lipman et al. (2022) have recently emerged as highly promising solutions to learn the underline data distribution in image generation, outperforming GAN-based models Brock et al. (2018); Sauer et al. (2022) and Auto-Regressive models Chang et al. (2022) by a significant margin.

However, diffusion models necessitate numerous denoising steps during inference, which incur a substantial computational cost, thereby limiting the widespread deployment of pre-trained diffusion models. To achieve fast diffusion sampling, the existing studies have explored two distinct approaches. Training-based techniques involve distilling the fast ODE trajectory into the model parameters, thereby circumventing redundant refinement steps. In addition, solver-based methods Lu et al. (2023); Zhang & Chen (2023); Song et al. (2020a) tackle the fast sampling problem by designing high-order numerical ODE solvers.

For training-based acceleration, Salimans & Ho (2022) aligns the single-step student denoiser with the multi-step teacher output, thereby reducing inference burdens. The consistency model concept, introduced by Song et al. (2023), directly teaches the model to produce consistent predictions at any arbitrary timesteps. Building upon Song et al. (2023), subsequent works Zheng et al. (2024); Kim et al. (2023); Wang et al. (2024); Xu et al. (2024) propose improved techniques to mitigate discreet errors in LCM training. Furthermore, Lin et al. (2024); Kang et al. (2024); Yin et al. (2024); Zhou et al. (2024) leverage adversarial training and distribution matching to enhance the quality of generated samples. To improve the training efficiency of distribution matching. However, training-based methods introduce changes to the model parameters, resulting in an inability to fully exploit the pre-training performance.

Solver-based methods rely heavily on the ODE formulation in the reverse-diffusion dynamics and hand-crafted multi-step solvers. Lu et al. (2023; 2022) and Zhang & Chen (2023) point out the semi-linear structure of the diffusion ODE and propose an exponential integrator to tackle faster sampling in diffusion models. Zhao et al. (2023) further enhances the sampling quality by borrowing the predictor-corrector structure. Thanks to the multistep-based ODE solver methods, high-quality samples can be generated within as few as 10 steps. To further improve efficiency, Gao et al. (2023) tracks the backward error and determines the adaptive step. Moreover, Karras et al. (2022); Lu et al. (2022) propose a handcrafted timesteps scheduler to sample respaced timesteps. Xue et al. (2024) argues that timesteps sampled in Karras et al. (2022); Lu et al. (2022) are suboptimal, thus proposing an online optimization algorithm to find the optimal sampling timesteps for generation. Apart from timesteps optimization, Shaul et al. (2023) learns a specific path transition to improve the sampling efficiency.

In contrast to training-based acceleration methods, solver-based approaches do not necessitate parameter adjustments and preserve the optimal performance of the pre-trained model. Moreover, solvers can be seamlessly applied to any arbitrary diffusion model trained with a similar noise scheduler, offering a high degree of flexibility and adaptability. This motivates us to investigate the generative capabilities of pre-trained diffusion models within limited steps from a diffusion solver perspective.

Current state-of-the-art diffusion solvers Lu et al. (2023); Zhao et al. (2023) adopt Adams-like multi-step methods that use the Lagrange interpolation function to minimize integral errors. We argue that an optimal solver should be tailored to specific pre-trained denoising functions and their corresponding noise schedulers. In this paper, we explore solver-based methods for fast diffusion sampling by improving diffusion solvers using data-driven approaches without destroying the pre-training internality in diffusion models. Inspired by Xue et al. (2024), we investigate the sources of error in the diffusion ODE and discover that the interpolation function form is inconsequential and can be reduced to coefficients. Furthermore, we define a compact search space related to the timesteps and solver coefficients. Therefore, we propose a differentiable solver search method to identify the optimal parameters in the compact search space.

Based on our novel differentiable solver search algorithm, we investigate the upper bound performance of pre-trained diffusion models under limited steps. Our searched solver significantly improves the performance of pre-trained diffusion models, and outperforms traditional solvers with a large gap. On ImageNet-256 $\times$ 256, armed with our solver, rectified-flow models of SiT-XL/2 and FlowDCN-XL/2 achieve 2.40 and 2.35 FID respectively under 10 steps, while DDPM model of DiT-XL/2 achieves 2.33 FID. Surprisingly, our findings reveal that when equipped with an optimized high-order solver, the DDPM can achieve comparable or even surpass the performance of rectified flow models under similar step constraints.

To summarize, our contributions are

- We reveal that the interpolation function choice is not important and can be reduced to coefficients through the pre-integral technique. We demonstrate that the upper bound of discretization error in reverse-diffusion ODE is related to both timesteps and solver coefficients and define a compact solver search space.

- Based on our analysis, we propose a novel differentiable solver search algorithm to find the optimal solver parameter for given diffusion models.

- For DDPM/VP time scheduling, armed with our searched solver, DiT-XL/2 achieves 2.33 FID under 10 steps, beating DPMSolver++/UniPC by a significant margin.

- For Rectified-flow models, armed with our searched solver, SiT-XL/2 and FlowDCN-XL/2 achieve 2.40 and 2.35 FID respectively under 10 steps on ImageNet-$256 \times 256$.
- For Text-to-Image diffusion models like FLUX, SD3, PixArt-$\Sigma$, our solver searched on ImageNet-$256 \times 256$ consistently yields better images compared to traditional solvers with the same CFG scale.

## 2 RELATED WORKS

**Diffusion Model** gradually adds $\boldsymbol{x}_0$ with Gaussian noise $\epsilon$ to perturb the corresponding known data distribution $p(x_0)$ into a simple Gaussian distribution. The discrete perturbation function of each $t$ satisfies $\mathcal{N}(\boldsymbol{x}_t | \alpha_t \boldsymbol{x}_0, \sigma_t^2 \boldsymbol{I})$, where $\alpha_t, \sigma_t > 0$. It can also be written as Equation (1).

$$\boldsymbol{x}_t = \alpha_t \boldsymbol{x}_{\text{real}} + \sigma_t \boldsymbol{\epsilon} \tag{1}$$

Moreover, as shown in Equation (2), Equation (1) has a forward continuous-SDE description, where $f(t) = \frac{\mathrm{d}\log \alpha_t}{\mathrm{d}t}$ and $g(t) = \frac{\mathrm{d}\sigma_t^2}{\mathrm{d}t} - \frac{\mathrm{d}\log \alpha_t}{\mathrm{d}t}\sigma_t^2$. Anderson (1982) establishes a pivotal theorem that the forward SDE has an equivalent reverse-time diffusion process as in Equation (3), so the generating process is equivalent to solving the diffusion SDE. Typically, diffusion models employ neural networks and distinct prediction parametrization to estimate the score function $\nabla \log_x p_{\boldsymbol{x}_t}(\boldsymbol{x}_t)$ along the sampling trajectory Song et al. (2020b); Karras et al. (2022); Ho et al. (2020).

$$d\boldsymbol{x}_t = f(t)\boldsymbol{x}_t \mathrm{d}t + g(t)\mathrm{d}\boldsymbol{w} \tag{2}$$

$$d\boldsymbol{x}_t = [f(t)\boldsymbol{x}_t - g(t)^2 \nabla_{\boldsymbol{x}} \log p(\boldsymbol{x}_t)]dt + g(t)d\boldsymbol{w} \tag{3}$$

Song et al. (2020b) also shows that there exists a corresponding deterministic process Equation (4) whose trajectories share the same marginal probability densities of Equation (3).

$$d\boldsymbol{x}_t = [f(t)\boldsymbol{x}_t - \frac{1}{2}g(t)^2 \nabla_{\boldsymbol{x}} \log p(\boldsymbol{x}_t)]dt \tag{4}$$

**Rectified Flow Model** simplifies diffusion model under the framework of Equation (2) and Equation (3). Different from Ho et al. (2020) introduces non-linear transition scheduling, the rectified-flow model adopts linear function to transform data to standard Gaussian noise.

$$\boldsymbol{x}_t = t\boldsymbol{x}_{\text{real}} + (1-t)\boldsymbol{\epsilon} \tag{5}$$

Instead of estimating the score function $\nabla \log_{\boldsymbol{x}_t} p_t(\boldsymbol{x}_t)$, rectified-flow models directly learn a neural network $v_\theta(x_t, t)$ to predict the velocity field $\boldsymbol{v}_t = d\boldsymbol{x}_t = (\boldsymbol{x}_{\text{real}} - \boldsymbol{\epsilon})$.

$$\mathcal{L}(\theta) = \mathbb{E}[\int_0^1 ||\boldsymbol{v}_\theta(\boldsymbol{x}_t, t) - \boldsymbol{v}_t||^2 \mathrm{d}t] \tag{6}$$

**Solver-based Fast Sampling Method** does not necessitate parameter adjustments and preserves the optimal performance of the pre-trained model. It can be seamlessly applied to an arbitrary diffusion model trained with a similar noise scheduler, offering a high degree of flexibility and adaptability. Solvers heavily rely on the reverse diffusion ODE in Equation (4). Current solvers are mainly focused on DDPM/VP noise schedules. Lu et al. (2022); Zhang & Chen (2023) discovered the semi-linear structure in DDPM/VP reverse ODEs. Furthermore, Zhao et al. (2023) enhanced the sampling quality by borrowing the predictor-corrector structure. Thanks to the multi-step ODE solvers, high-quality samples can be generated within as few as 10 steps. To further improve efficiency, Gao et al. (2023) tracks the backward error and determines the adaptive step. Moreover, Karras et al. (2022); Lu et al. (2022) proposed a handcrafted timestep scheduler to sample respaced timesteps. However, Xue et al. (2024) argued that the timestep sampled in Karras et al. (2022); Lu et al. (2022) is suboptimal, and thus proposed an online optimization algorithm to find the optimal sampling timestep for generation. Apart from timestep optimization, Shaul et al. (2023) learned a specific path transition to improve the sampling efficiency.

## 3 PROBLEM DEFINITION

As rectified-flow constitutes a simple yet elegant formulation within the diffusion family, we choose rectified-flow as the primary subject of discussion in this paper to enhance readability. Importantly,

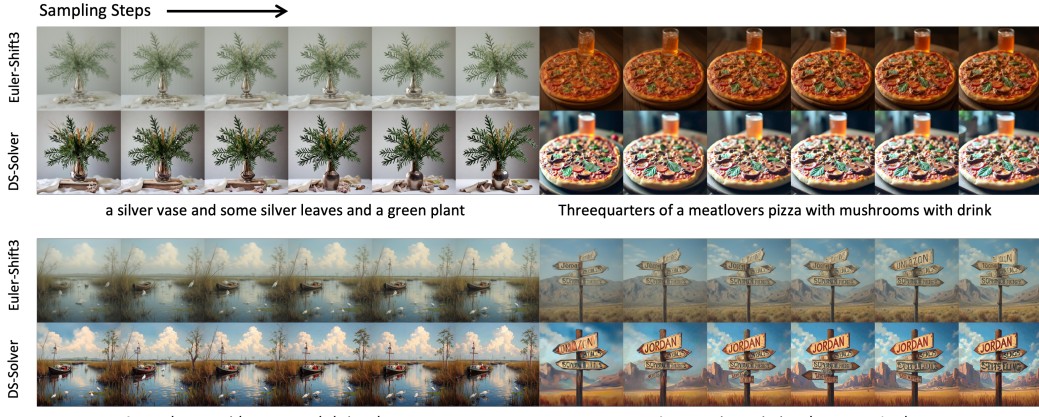

Figure 2: **Generated images from Flux.1-dev with Guidance=2.0 and our solver (searched on SiT-XL/2).** Euler-Shift3 is the default solver provided by diffusers and Flux community. Our solver(DS-Solver) achieves better visual quality from 5 to 10 steps(NFE).

our proposed algorithm is not constrained to rectified-flow models. We explore its applicability to other diffusion models such as DDPM/VP in Section 6.

Recall the continuous integration of reverse-diffusion in Equation (7) with the pre-defined interval $\{t_0, t_1, ...t_N\}$. Given the pre-trained diffusion models and their corresponding ODE defined in Equation (4), before we tackle the integration of interval $[t_i, t_{i+1}]$, we have already obtained the sampled velocity field of previous timestep $\{(\boldsymbol{x}_j, t_j, \boldsymbol{v}_j = \boldsymbol{v}_\theta(\boldsymbol{x}_j, t_j))\}_{j=0}^i$. Here, we directly denote $\boldsymbol{x}_{t_i}$ as $\boldsymbol{x}_i$ for presentation clarity:

$$\boldsymbol{x}_{i+1} = \boldsymbol{x}_i + \int_{t_i}^{t_{i+1}} \boldsymbol{v}_\theta(\boldsymbol{x}_t, t)dt \tag{7}$$

As shown in Equation (8), we strive to develop **a more optimal solver** that minimizes the integral error while enhancing image quality under limited sampling steps (NFE) without requiring any parameter adjustments for the pre-trained model.

$$\Phi = \arg\min \mathbb{E}[||\Phi(\boldsymbol{\epsilon}, \boldsymbol{v}_\theta) - (\boldsymbol{\epsilon} + \int_0^1 \boldsymbol{v}_\theta(\boldsymbol{x}_t, t)dt)||]. \tag{8}$$

## 4 ANALYSIS OF REVERSE-DIFFUSION ODE SAMPLING

Initially, we revisit the multi-step methods commonly used by Zhao et al. (2023); Zhang & Chen (2023); Lu et al. (2023) and identify potential limitations. Specifically, we argue that the Lagrange interpolation function used in Adams-Bashforth methods is suboptimal for diffusion models. Moreover, we show that the specific form of the interpolation function is inconsequential, as pre-integration and expectation estimation ultimately reduce it to a set of coefficients. Inspired by Xue et al. (2024), we prove that timesteps and these coefficients effectively constitute our search space.

### 4.1 RECAP THE MULTI-STEP METHODS

As shown in Equation (9), the Euler method employs $\boldsymbol{v}_i$ as the estimation of Equation (9) in whole interval $[t_i, t_{i+1}]$. Higher-order multi-step solvers further improve the estimation quality of the integral by incorporating interpolation functions and leveraging previously sampled values.

$$\boldsymbol{x}_{i+1} = \boldsymbol{x}_i + (t_{i+1} - t_i)\boldsymbol{v}_\theta(\boldsymbol{x}_i, t_i). \tag{9}$$

The most classic multi-step solver Adams–Bashforth method Bashforth & Adams (1883) incorporates the Lagrange polynomial to improve the estimation accuracy within a given interval. It is noteworthy that the number of NFE and sampling steps are essentially the same for multi-step methods. In contrast, Runge-Kutta and Huen methods require more NFE for a given number of sampling

steps.

$$\boldsymbol{x}_{i+1} \approx \boldsymbol{x}_i + \int_{t_i}^{t_{i+1}} \sum_{j=0}^{i} (\prod_{k=0, k \neq j}^{i} \frac{t - t_k}{t_j - t_k}) \boldsymbol{v}_j dt \tag{10}$$

$$\boldsymbol{x}_{i+1} \approx \boldsymbol{x}_i + \sum_{j=0}^{i} \boldsymbol{v}_j \int_{t_i}^{t_{i+1}} (\prod_{k=0, k \neq j}^{i} \frac{t - t_k}{t_j - t_k}) dt \tag{11}$$

As Equation (11) states, $\int_{t_i}^{t_{i+1}} (\prod_{k=0, k \neq j}^{i} \frac{t - t_k}{t_j - t_k}) dt$ of the Lagrange polynomial can be pre-integrated into a constant coefficient, resulting in only naive summation being required for ODE solving. Current SoTA multi-step solvers Lu et al. (2023); Zhao et al. (2023) are heavily inspired by Adams–Bashforth-like multi-step solvers. These solvers employ the Lagrange interpolation function or difference formula to estimate the value in the given interval.

However, the Lagrange interpolation function and other similar methods only take $t$ into account while the $\boldsymbol{v}(\boldsymbol{x}, t)$ also needs $\boldsymbol{x}$ as inputs. Using first-order Taylor expansion of $\boldsymbol{x}$ at $\boldsymbol{x}_i$ and higher-order expansion of $t$ at $t_i$, we can readily derive the error bound of the estimation.

## 4.2 Focus on Solver coefficients instead of the interpolation function

Different from general ODE solving problems, a compact searching space exists given reverse-diffusion ODE and pre-trained models. We define a universal interpolation function $\mathcal{P}$ without an explicit form. $\mathcal{P}$ measures the distance of $(\boldsymbol{x}_t, t)$ between previous sampled points $\{(\boldsymbol{x}_j, t_j)\}_{j=0}^{i}$ to determine the interpolation weight for $\{\boldsymbol{v}_j\}_{j=0}^{i}$.

$$\boldsymbol{x}_{i+1} \approx \boldsymbol{x}_i + \int_{t_i}^{t_{i+1}} \sum_{j=0}^{i} \mathcal{P}(\boldsymbol{x}_t, t, \boldsymbol{x}_j, t_j) \boldsymbol{v}_j dt. \tag{12}$$

**Assumption 4.1.** We assume that the remainder term of the universal interpolation function $\sum_{j=0}^{i} \mathcal{P}(\boldsymbol{x}_t, t, \boldsymbol{x}_j, t_j) \boldsymbol{v}_j$ for $v(\boldsymbol{x}, t)$ is bound as $\mathcal{O}(d\boldsymbol{x}^m) + \mathcal{O}(dt^n)$, where $\mathcal{O}(d\boldsymbol{x}^m)$ is the $m$th-order infinitesimal for $d\boldsymbol{x}$, $\mathcal{O}(dt^m)$ is the $n$th-order infinitesimal for $dt$.

Equation (12) has a recurrent dependency, as $\boldsymbol{x}_t$ also relies on $\sum_{j=0}^{i} \mathcal{P}(\boldsymbol{x}_t, t, \boldsymbol{x}_j, t_j) \boldsymbol{v}_j dt$. To eliminate the recurrent dependency, shown in Equation (13), we simply use the first order Taylor expansion of $x(t)$ at $x_i$ to replace the original form. Recall that $\boldsymbol{v}_i$ is already determined by $\boldsymbol{x}_i$ and $t_i$, thus the partial integral of Equation (13) can be formulated as Equation (14). Different from the naive Lagrange interpolation, $\mathcal{C}_j(\boldsymbol{x}_i)$ is a function of current $\boldsymbol{x}_i$ instead of a constant scalar. Learning a $\mathcal{C}_j(\boldsymbol{x}_i)$ function will cause the generalization to be lost. This limits the actual usage in diffusion model sampling.

$$\boldsymbol{x}_{i+1} \approx \boldsymbol{x}_i + \sum_{j=0}^{i} \boldsymbol{v}_j \int_{t_i}^{t_{i+1}} \mathcal{P}(\boldsymbol{x}_i + \boldsymbol{v}_i(t - t_i), t, \boldsymbol{x}_j, t_j) dt \tag{13}$$

$$\boldsymbol{x}_{i+1} \approx \boldsymbol{x}_i + \sum_{j=0}^{i} \boldsymbol{v}_j \mathcal{C}_j(\boldsymbol{x}_i)(t_{i+1} - t_i) \tag{14}$$

**Theorem 4.2.** *Given sampling time interval $[t_i, t_{i+1}]$ and suppose $\mathcal{C}_j(\boldsymbol{x}_i) = g_j(\boldsymbol{x}_i) + b_i^j$, Adams-like linear multi-step methods have an error expectation of $(t_{i+1} - t_i)\mathbb{E}_{\boldsymbol{x}_i}||\sum_{j=0}^{i} \boldsymbol{v}_j g_j(\boldsymbol{x}_i)||$. replacing $\mathcal{C}_j(\boldsymbol{x})$ with $\mathbb{E}_{\boldsymbol{x}_i}[\mathcal{C}_j(\boldsymbol{x}_i)]$ is the optimal choice and owns an error expectation of $(t_{i+1} - t_i)\mathbb{E}_{\boldsymbol{x}_i}||\sum_{j=0}^{i} \boldsymbol{v}_j[g_j(\boldsymbol{x}_i) - \mathbb{E}_{\boldsymbol{x}_i}g_j(\boldsymbol{x}_i)||$. We place the proof in Appendix A.*

According to Theorem 4.2, we opt to replace $\mathcal{C}_j(\boldsymbol{x}_i)$ with its expectation $\mathbb{E}_{\boldsymbol{x}_i}[\mathcal{C}_j(\boldsymbol{x}_i)]$, thus we obtain diffusion-scheduler related coefficients while keeping generalization ability. Finally, given the pre-defined time intervals, we obtain the optimization target Equation (15), where $c_i^j = \mathbb{E}_{\boldsymbol{x}_i}[\mathcal{C}_j(\boldsymbol{x}_i)]$. The expectation can be deemed as optimized through massive data and gradient descent.

$$\boldsymbol{x}_{i+1} \approx \boldsymbol{x}_i + \sum_{j=0}^{i} \boldsymbol{v}_j c_i^j (t_{i+1} - t_i) \tag{15}$$

Figure 3: **Generated images from SD3 with CFG=4.0 and our solver (searched on SiT-XL/2).** Euler-Shift3 is the default solver provided by diffusers and SD3 community. Our solver achieves better visual quality in from 8 to 10 steps(NFE).

### 4.3 OPTIMAL SEARCH SPACE FOR A SOLVER

**Assumption 4.3.** As shown in Equation (16), the pre-trained velocity model $\boldsymbol{v}_\theta$ is not perfect and the error between $\boldsymbol{v}_\theta$ and ideal velocity field $\hat{\boldsymbol{v}}$ is L1-bounded, where $\eta$ is a constant scalar.

$$||\hat{\boldsymbol{v}} - \boldsymbol{v}_\theta|| \leq \eta \ll ||\hat{\boldsymbol{v}}|| \tag{16}$$

Previous discussions assume we have a perfect velocity function. However, the ideal velocity is hard to obtain, we only have pre-trained velocity models. Following Equation (15), we can expand Equation (15) from $t_{i=0}$ to $t_{i=N}$ to obtain the error bound caused by non-ideal velocity estimation.

**Theorem 4.4.** *The error caused by the non-ideal velocity estimation model can be formulated in the following equation. We can employ triangle inequalities to obtain the error-bound(L1) of* $||\boldsymbol{x}_N - \hat{x}_N||$, *the proof can be found in the Appendix B.*

$$||\boldsymbol{x}_N - \hat{x}_N|| \leq \eta \sum_{i=0}^{N-1} \sum_{j=0}^{i} |c_i^j(t_{i+1} - t_i)|$$

Based on Theorem 4.4, since the error bound is related to timesteps and solver coefficients, we can define a much more compact search space consisting of $\{c_i^j\}_{j<i,j=0,i=1}^N$ and $\{t_i\}_{i=0}^N$.

**Theorem 4.5.** *Based on Theorem 4.4 and Theorem 4.2. We can derive the total upper error bound(L1) of our solver search method and other counterparts. The total upper error bound of Our solver search is:*

$$\sum_{i=0}^{N-1} (t_{i+1} - t_i)(\sum_{j=0}^{i} \eta |\mathbb{E}_{\boldsymbol{x}_i} g_j(\boldsymbol{x}_i) + b_i^j| + \mathbb{E}_{\boldsymbol{x}_i} ||\sum_{j=0}^{i} \boldsymbol{v}_j g_j(\boldsymbol{x}_i) - \mathbb{E}_{\boldsymbol{x}_i} g_j(\boldsymbol{x}_i)||)$$

*Compared to Adams-like linear multi-step methods. Our searched solver has a small upper error bound. The proof can be found in the Appendix B.*

Through Theorem 4.5, our searched solvers own a relatively small upper error bound. Thus we can theoretically guarantee optimal compared to Adams-like methods.

## 5 DIFFERENTIABLE SOLVER SEARCH.

Through previous discussion and analysis, we identify $\{c_i^j\}_{j<i,j=0,i=1}^N$ and $\{t_i\}_{i=0}^N$ as the target search items. To this end, we propose a data-driven, differentiable solver search approach to determine these target items.

---

**Algorithm 1** Solver Parametrization

**Requires:** $\{r_i, \}$ and $\{c_i^j, \}$
**TimeDeltas:** $\Delta t_0, \Delta t_1, ..., \Delta t_{n-1}$.
**SolverCoefficients:** $\mathcal{M} \in R^{N \times N}$
$\{\Delta t_i, \} = \text{Softmax}(\{r_i\})$

$$\mathcal{M} = \begin{bmatrix} 1 & & & \\ c_1^0 & 1 - c_1^0 & & \\ \vdots & \vdots & \vdots & \ddots \\ c_{n-1}^0 & c_{n-1}^1 & \cdots & 1 - \sum_{k=0}^{n-1} c_{n-1}^k \end{bmatrix}$$

---

**Algorithm 2** Differentiable Solver Search

**Require:** $\boldsymbol{v}_\theta$ model, $\{\Delta t_i, \}_{i=0}^{N-1}$, $\mathcal{M}$, A buffer $Q$.
Compute $\{\tilde{\boldsymbol{x}}_l, \}_{l=0}^L = \textbf{Euler}(\boldsymbol{\epsilon}, v_\theta)$.
**for** $i = 0$ **to** $N - 1$ **do**
$\quad Q \overset{\text{buffer}}{\longleftarrow} \boldsymbol{v}_\theta(\boldsymbol{x}_{t_i}, t_i)$
$\quad$ Compute $\boldsymbol{v} = \sum_{j=0}^i \mathcal{M}_{ij} Q_j$.
$\quad t_{i+1} = t_i + \Delta t_i$
$\quad \boldsymbol{x}_{t_{i+1}} = \boldsymbol{x}_{t_i} + \boldsymbol{v} \Delta t_i$
**end for**
**return:** $\tilde{\boldsymbol{x}}_{t_{n-1}}, \mathcal{L}(\{\tilde{\boldsymbol{x}}_l\}_{l=0}^L, \{\boldsymbol{x}_i\}_{i=0}^N)$

---

**Timestep Parametrization.** As shown in Algorithm 1, we employ unbounded parameters $\{r_i, \}_{i=0}^{N-1}$ as the optimization subject, as the integral interval is from 0 to 1, we convert $r_i$ into time-space deltas $\Delta t_i$ with softmax normalization function to force their summation to 1. We can access timestep $t_{i+1}$ through $t_{i+1} = t_i + \Delta t_i$. We initialize $\{r_i\}_{i=0}^{N-1}$ with 1.0 to obtain a uniform timestep distribution.

**Coefficients Parametrization.** Inspired by Xue et al. (2024). Given Equation (15) and Equation (7), when the velocity field $v_\theta(x, t)$ yields constant value, an implicit constraint $\sum_{k=0}^i c_k^i = 1$ emerges. This observation motivates us to re-parameterize the diagonal value of $M$ as $\{1 - \sum_{j=0}^{i-1} c_i^j, \}_{i=0}^{N-1}$. We initialize $\{c_i^k, \}$ with zeros to mimic the behavior of the Euler solver.

**Mono-alignment Supervision.** We take the $L$-step Euler solver's ODE trajectory $\{\tilde{\boldsymbol{x}}\}_{l=0}^L$ as reference. We minimize the gap between the target and source trajectories with the MSE loss. We also adopt Huber loss as auxiliary supervision for $\boldsymbol{x}_{t_N}$.

## 6 EXTENDING TO DDPM/VP FRAMEWORK

Applying our differentiable solver search to DDPM is infeasible. However, Song et al. (2020b) suggests that there exists a continuous SDE process with $\{f(t) = -\frac{1}{2}\beta_t; g(t) = \sqrt{\beta_t}\}$ corresponding to discrete DDPM. This motivates us to transform the search space from the infeasible discrete space to its continuous SDE counterpart. Lu et al. (2022) and Zhang & Chen (2023) discover the semi-linear structure of diffusion and propose exponential integral with $\epsilon$ parametrization to tackle the fast sampling problem of DDPM models, where $\alpha_t = e^{\int_0^t -\frac{1}{2}\beta_s ds}$, $\sigma_t = \sqrt{1 - e^{\int_0^t -\beta_s ds}}$ and $\lambda_t = \log \frac{\alpha_t}{\sigma_t}$. Lu et al. (2023) further discovers that $x$ parametrization is more powerful for diffusion sampling under limited steps, where $\bar{\boldsymbol{x}} = \frac{\boldsymbol{x}_t - \sigma\boldsymbol{\epsilon}}{\alpha_t}$.

$$\boldsymbol{x}_t = \frac{\sigma_t}{\sigma_s}\boldsymbol{x}_s + \sigma_t \int_{\lambda_s}^{\lambda_t} e^\lambda \bar{\boldsymbol{x}}_\theta(\boldsymbol{x}_{t(\lambda)}, t(\lambda)) \mathrm{d}\lambda \tag{17}$$

We opt to follow the $\bar{x}$ parametrization as DPM-Solver++. However, we find directly interpolating $e^\lambda \boldsymbol{x}_\theta(x_t, t)$ as a whole part is hard for searching, and yields worse results. To avoid conflating the interpolation coefficients with exponential integral, we employ $\omega_t = \frac{\alpha_t}{\sigma_t}$ and transform Equation (17) into Equation (18) with a similar interpolation format as Equation (14), where $t(\omega)$ maps $\omega$ to timestep.

$$\boldsymbol{x}_t \approx \frac{\sigma_t}{\sigma_s}\bar{\boldsymbol{x}}_s + \sigma_t(\omega_t - \omega_s) \sum_{k=1}^i c_i^k \boldsymbol{x}_\theta(\bar{\boldsymbol{x}}_k, t_k) \tag{18}$$

## 7 EXPERIMENT

We demonstrate the efficiency of our differentiable solver search by conducting experiments on publicly available diffusion models. Specifically, we utilize DiT-XL/2 Peebles & Xie (2023) trained with DDPM scheduling and rectified-flow models SiT-XL/2 Ma et al. (2024) and FlowDCN-XL/2 Anonymous (2024). Our default training setting employs the Lion optimizer Chen et al.

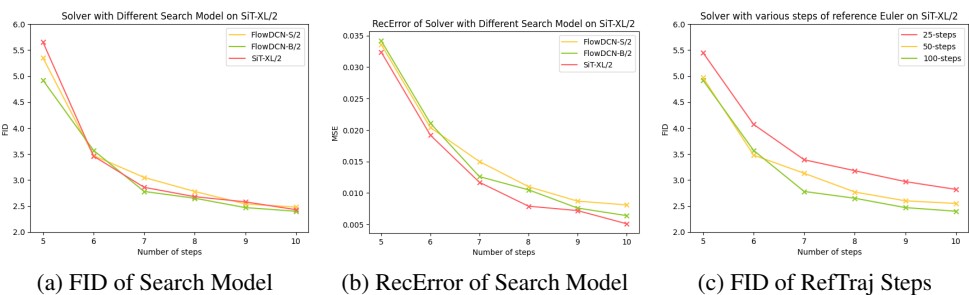

(a) FID of Search Model     (b) RecError of Search Model     (c) FID of RefTraj Steps

Figure 4: **Ablations studies of Differentiable Solver Search.** We evaluate the searched solver on SiT-XL/2, and report the FID performance curve of searched solvers.

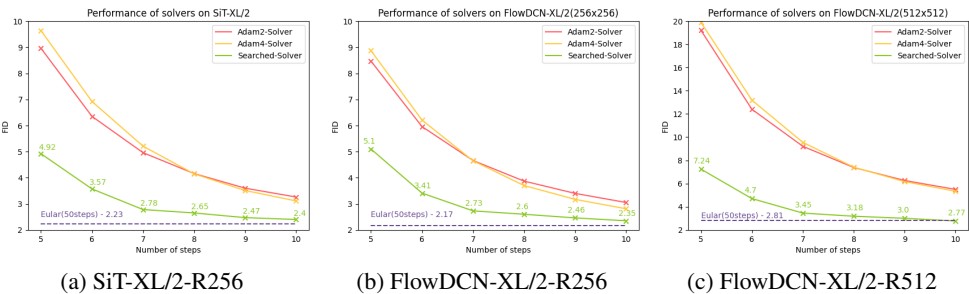

(a) SiT-XL/2-R256     (b) FlowDCN-XL/2-R256     (c) FlowDCN-XL/2-R512

Figure 5: **The same searched solver on different Rectified-Flow Models.** R256 and R512 indicate the generation resolution of given model. We search solver with FlowDCN-B/2 on ImageNet-256 × 256 and evaluate it with SiT-XL/2 and FlowDCN-XL/2 on different resolution datasets. Our searched solver outperforms traditional solvers by a significant margin. More metrics(sFID, IS, Precision, Recall) are places at Appendix

(2024b) with a constant learning rate of 0.01 and no weight decay. We sample 50,000 images for the entire search process. Notably, searching with 50,000 samples using FlowDCN-B/2 requires approximately 30 minutes on 8 × H20 computation cards. During the search, we deliberately avoid using CFG to construct reference and source trajectories, thereby preventing misalignment.

## 7.1 RECTIFIED FLOW MODELS

We search solver with FlowDCN-B/2, FlowDCN-S/2 and SiT-XL/2. We compare the search solver's performance with the second-order and fourth-order Adam multi-step method on SiT-XL/2, FlowDCN-XL/2 trained on 256 × 256 and FlowDCN-XL/2 trained on 512 × 512.

**Search Model.** We tried different search models among different size and architecture. We report the FID performance and reconstruction error of SiT-XL/2 in Figure 4a and Figure 4b respectively. Surprisingly, we find that the FID performance of SiT-XL/2 equipped with the solver searched using FlowDCN-B/2 outperforms the solver searched on SiT-XL/2 itself. Meanwhile, the reconstruction error between the sampled result produced by Euler-250 steps is as expected. These findings suggest that there exists a minor discrepancy between FID and the pursuit of minimal error in the current solver design.

**Step of Reference Trajectory.** We provide reference trajectory $\{\tilde{x}\}_{l=0}^{L}$ of different sampling step $L$ for differentiable solver search. We take FlowDCN-B/2 as the search model and report the FID measured on SiT-XL/2 in Figure 4c. As the sampling step of reference trajectory increases, the FID of SiT-XL/2 further improves and becomes better. However, the performance improvement is not significant at 5 and 6 steps, suggesting that the improvement bound for extremely limited steps.

**ImageNet** 256 × 256**.** We validate the searched solver on SiT-XL/2 and FlowDCN-XL/2. We arm the pre-trained model with CFG of 1.375. As shown in Figure 5a, our searched solver improves FID performance significantly and achieves 2.40 FID under 10 steps. As shown in Figure 5b, our searched solver achieves 2.35 FID under 10 steps, beating traditional solvers by large margins.

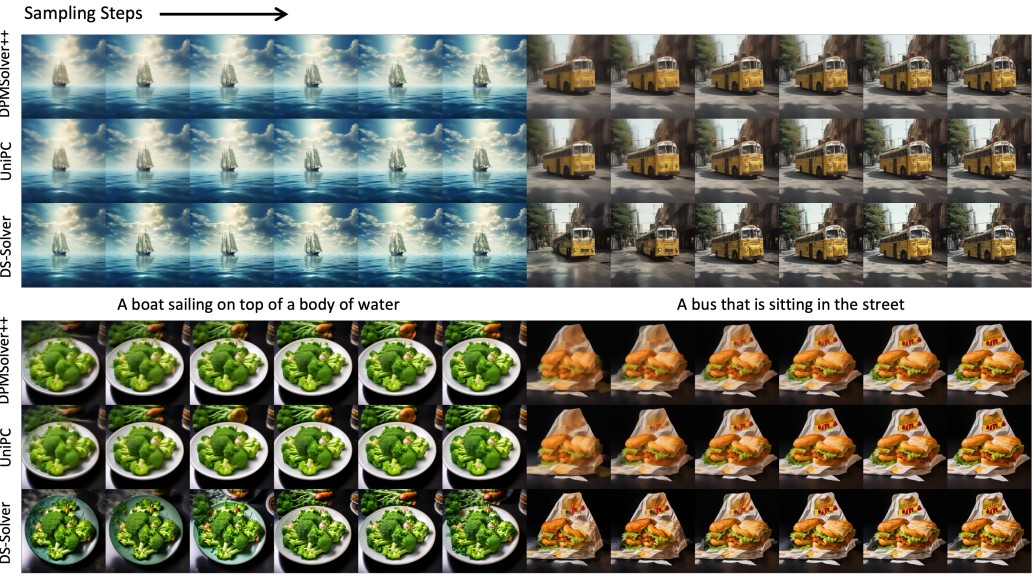

Figure 6: The images generated from PixArt-$\Sigma$ with CFG=2.0 equipped with Our DS-Solver ( searched on DiT-XL/2-R256 ).In comparison to DPM-Solver++ and UniPC, our results consistently exhibit greater clarity and possess more details. Our solver (DS-Solver) achieves better quality from 5 to 10 steps(NFE).

**ImageNet** $512 \times 512$. Since Ma et al. (2024) has not released SiT-XL/2 trained on $512 \times 512$ resolution, we directly report the performance collected from FlowDCN-XL/2. We arm FlowDCN-XL/2 with CFG of 1.375 and four channels. Our searched solver achieves 2.77 FID under 10 steps, beating traditional solver by a large margin, even slightly outperforming the Euler solver with 50 steps(2.81FID).

**Text to Image.** Shown in Figure 2 and Figure 3, we apply our solver search on FlowDCN-B/2 and SiT-XL/2 to the most advanced Rectified-Flow model Flux.1-dev and SD3 Esser et al. (2024). We find Flux.1-Dev would produce grid points in generation. To alleviate the grid pattern, we decouple the velocity field into mean and direction, only apply our solver to the direction, and replace the mean with an exponential decayed mean. The details can be found in the appendix.

## 7.2 DDPM/VP Models

We choose the open-source model DiT-XL/2 trained on ImageNet $256 \times 256$ as the search model to conduct experiments. We compare the performance of the searched solver with DPM-Solver++ and UniPC on ImageNet $256 \times 256$ and ImageNet $512 \times 512$.

**ImageNet** $256 \times 256$. Following Peebles & Xie (2023) and Xue et al. (2024), We arm pre-trained DiT-XL/2 with CFG of 1.5 and apply CFG only on the first three channels. As shown in Table 1, our searched solver improves FID performance significantly and achieves 2.33 FID under 10 steps.

**ImageNet** $512 \times 512$. We directly apply the solver searched on $256 \times 256$ resolution to ImageNet $512 \times 512$. The result is also great to some extent, DiT-XL/2($512 \times 512$) achieves 3.64 FID under 10 steps, outperforming DPM-Solver++ and UniPC with a large gap.

**Text to Image.** As we search solver with DiT and its corresponding noise scheduler, so it is infeasible to apply our solver to other DDPM models with different $\beta_{\min}$ and $\beta_{\max}$. Fortunately, we find Chen et al. (2024a) and Chen et al. (2023) also employ the same $\beta_{\min}$ and $\beta_{\max}$ as DiT. So we can provide the visualization results of our searched solver on PixArt-$\Sigma$ and PixArt-$\alpha$. Our visualization result is produced with CFG of 2.

| Methods \NFEs | 5 | 6 | 7 | 8 | 9 | 10 |
|---|---|---|---|---|---|---|
| DPM-Solver++ with uniform-$\lambda$ Lu et al. (2023) | 38.04 | 20.96 | 14.69 | 11.09 | 8.32 | 6.47 |
| DPM-Solver++ with uniform-$t$ Lu et al. (2023) | 31.32 | 14.36 | 7.62 | 4.93 | 3.77 | 3.23 |
| DPM-Solver++ with uniform-$\lambda$-opt Xue et al. (2024) | 12.53 | 5.44 | 3.58 | 7.54 | 5.97 | 4.12 |
| DPM-Solver++ with uniform-$t$-opt Xue et al. (2024) | 12.53 | 5.44 | 3.89 | 3.81 | 3.13 | 2.79 |
| UniPC with uniform-$\lambda$ Zhao et al. (2023) | 41.89 | 30.51 | 19.72 | 12.94 | 8.49 | 6.13 |
| UniPC with uniform-$t$ Zhao et al. (2023) | 23.48 | 10.31 | 5.73 | 4.06 | 3.39 | 3.04 |
| UniPC with uniform-$\lambda$-opt Xue et al. (2024) | 8.66 | 4.46 | 3.57 | 3.72 | 3.40 | 3.01 |
| UniPC with uniform-$t$-opt Xue et al. (2024) | 8.66 | 4.46 | 3.74 | 3.29 | 3.01 | 2.74 |
| **Searched-Solver** | **7.40** | **3.94** | **2.79** | **2.51** | **2.37** | **2.33** |

Table 1: FID ($\downarrow$) of different NFEs on DiT-XL/2 (trained on ImageNet $256 \times 256$). *-opt* indicates online optimization of the timesteps scheduler.

| Methods \NFEs | 5 | 6 | 7 | 8 | 9 | 10 |
|---|---|---|---|---|---|---|
| UniPC with uniform-$\lambda$ Zhao et al. (2023) | 41.14 | 19.81 | 13.01 | 9.83 | 8.31 | 7.01 |
| UniPC with uniform-$t$ Xue et al. (2024) | 20.28 | 10.47 | 6.57 | 5.13 | 4.46 | 4.14 |
| UniPC with uniform-$\lambda$-opt Xue et al. (2024) | 11.40 | **5.95** | 4.82 | 4.68 | 6.93 | 6.01 |
| UniPC with uniform-$t$-opt Xue et al. (2024) | 11.40 | **5.95** | 4.64 | 4.36 | 4.05 | 3.81 |
| **Searched-solver**(searched on DiT-XL/2-R256) | **10.28** | 6.02 | **4.31** | **3.74** | **3.54** | 3.64 |

Table 2: FID ($\downarrow$) of different NFEs on DiT-XL/2 (trained on ImageNet 512x512).

### 7.3 VISUALIZATION OF SOLVER PARAMETERS

**Searched Coefficients** are visualized in Figure 1. The absolute value of searched coefficients corresponding to DDPM/VP shares a different pattern, coefficients in DDPM/VP are more concentrated on the diagonal while rectified-flow demonstrates a more flattened distribution. This indicates there exists a more curved sampling path in DDPM/VP compared to rectified-flow.

**Searched Timesteps** are visualized in Figure 1. Compared to DDPM/VP, rectified-flow models more focus on the more noisy region, exhibiting small time deltas at the beginning. We fit the searched timestep of different NFE with polynomials and provide the respacing curves in Equation (19) and Equation (20). $t \in [0, 1]$, and $t = 0$ indicates the most noisy timestep.

$$\text{Rectified-Flow}: -1.96t^4 + 3.51t^3 - 0.97t^2 + 0.43t - 0.003 \tag{19}$$

$$\text{DDPM/VP}: -2.73t^4 + 6.30t^3 - 4.744t^2 + 2.17t - 0.0002 \tag{20}$$

## 8 CONCLUSION

We find a compact solver search space and propose a novel differentiable solver search algorithm to identify the optimal solver. Our searched solver outperforms traditional solvers by a significant margin. Equipped with the searched solver, DDPM/VP and Rectified Flow models significantly improve under limited sampling steps. However, our proposed solver still has several limitations(See Appendix), which we plan to address in future work.

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

648
649
650
651
652
653
654
655
656
657
658
659
660
661
662
663
664
665
666
667
668
669
670
671
672
673
674
675
676
677
678
679
680
681
682
683
684
685
686
687
688
689
690
691
692
693
694
695
696
697
698
699
700
701

# REBUTTALS

## Q.1 MORE METRICS OF SEARCHED SOLVER

We adhere to the evaluation guidelines provided by ADM and DM-nonuniform, reporting only the FID as the standard metric in Figure 5a. To clarify, we do not report selective results on rectified flow models; we present sFID, IS, PR, and Recall metrics for SiT-XL(R256), FlowDCN-XL/2(R256), and FlowDCN-B/2(R256). Our solver searched on FlowDCN-B/2, consistently outperforms the handcrafted solvers across FID, sFID, IS, and Recall metrics.

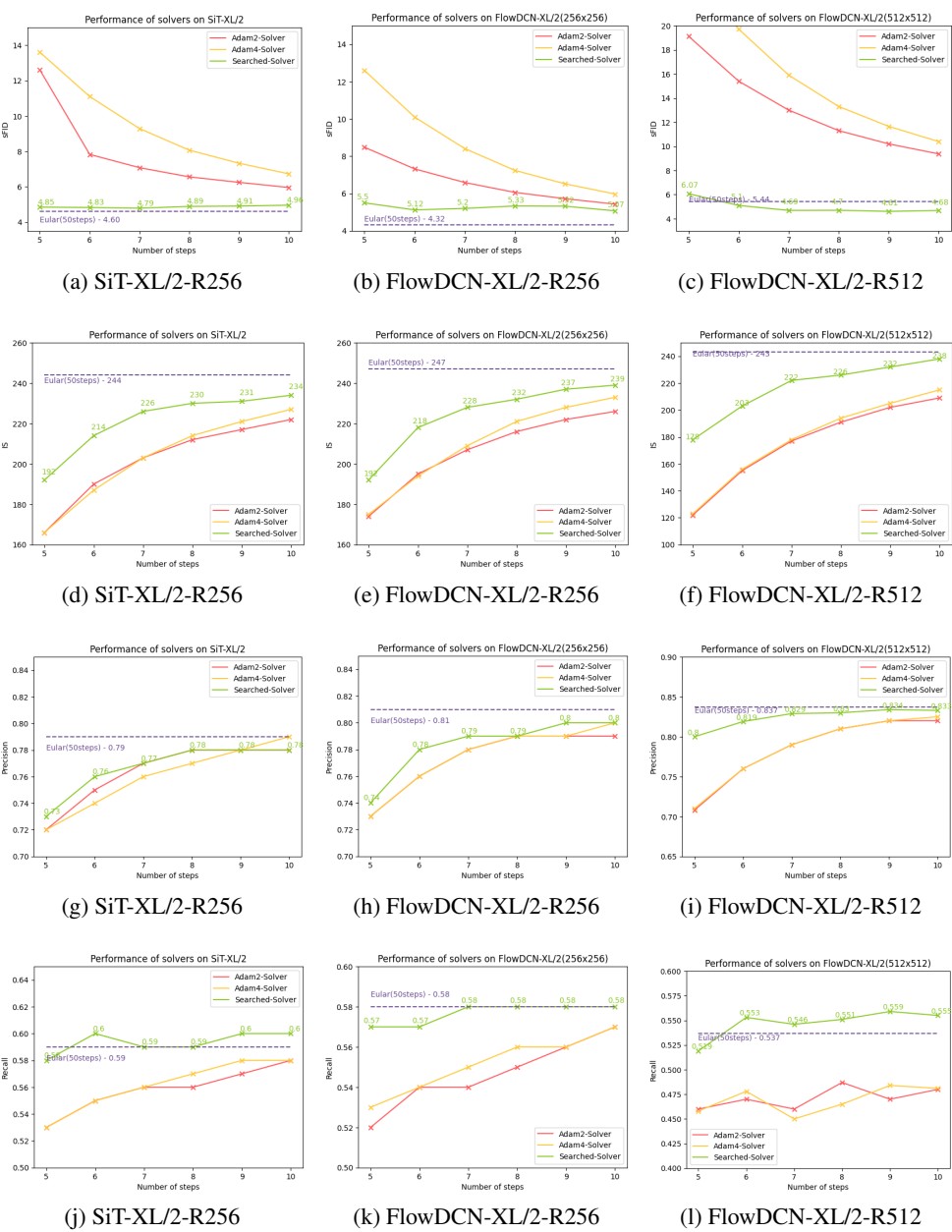

(a) SiT-XL/2-R256     (b) FlowDCN-XL/2-R256     (c) FlowDCN-XL/2-R512

(d) SiT-XL/2-R256     (e) FlowDCN-XL/2-R256     (f) FlowDCN-XL/2-R512

(g) SiT-XL/2-R256     (h) FlowDCN-XL/2-R256     (i) FlowDCN-XL/2-R512

(j) SiT-XL/2-R256     (k) FlowDCN-XL/2-R256     (l) FlowDCN-XL/2-R512

Figure 7: **The same searched solver on different Rectified-Flow Models.** R256 and R512 indicate the generation resolution of given model. We search solver with FlowDCN-B/2 on ImageNet-256 × 256 and evaluate it with SiT-XL/2 and FlowDCN-XL/2 on different resolution datasets. Our searched solver outperforms traditional solvers by a significant margin.

Q.2 10-STEP SOLVER OUTPERFORMING 50 EULER STEPS.

Linear multistep-based high-order solvers can significantly boost performance in simulations with a limited number of time steps. By leveraging the reference trajectory from the Euler solver with 100 steps, it is possible to outperform the Euler solver with 50 steps. As illustrated in all metrics, our solver enables SiT-XL/2-R256 and FlowDCN-XL/2-R256 to achieve better Recall scores than the Euler solver with 50 steps. Notably, FlowDCN-XL/2-R512 with our solver surpasses its Euler counterpart in terms of sFID, Precision, and Recall, demonstrating its exceptional performance.

Q.3 COMPUTATIONAL COMPLEXITY COMPARED TO OTHER METHODS.

**For sampling.** When performing sampling over $n$ time steps, our solver caches all pre-sampled predictions, resulting in a memory complexity of $\mathcal{O}(n)$. The model function evaluation also has a complexity of $\mathcal{O}(n)$ ($\mathcal{O}(2 \times n)$ for CFG enabled). It is important to note that the memory required for caching predictions is negligible compared to that used by model weights and activations. Besides classic methods, we also included a comparison with the latest Flowturbo published on NeurIPS24.

|  | Steps | NFE | NFE-CFG | Cache Pred | Order | search samples |
|---|---|---|---|---|---|---|
| Adam2 | n | n | 2n | 2 | 2 | / |
| Adam4 | n | n | 2n | 4 | 4 | / |
| heun | n | 2n | 4n | 2 | 2 | / |
| DPM-Solver++ | n | n | 2n | 2 | 2 | / |
| UniPC | n | n | 2n | 3 | 3 | / |
| FlowTurbo | n | >n | >2n | 2 | 2 | 540000(Real) |
| our | n | n | 2n | n | n | 50000(Generated) |

**For searching.** Solver-based algorithms, limited by their searchable parameter sizes, demonstrate significantly lower performance in few-step settings compared to distillation-based algorithms(5/6steps), making direct comparisons inappropriate. Consequently, we selected algorithms that are both acceleratable on ImageNet and comparable in performance, including popular methods such as DPM-Solver++, UniPC, and classic Adams-like linear multi-step methods. Since our experiments primarily utilize SiT, DiT, and FlowDCN that trained on the ImageNet dataset. We also provide fair comparisons by incorporating the latest acceleration method, FlowTurbo. Additionally, we have included results from the heun method as reported in FlowTurbo.

| SiT-XL-R256 | Steps | NFE-CFG | Extra-Paramters | FID | IS | PR | Recall |
|---|---|---|---|---|---|---|---|
| Heun | 8 | 16x2 | 0 | 3.68 | / | / | / |
| Heun | 11 | 22x2 | 0 | 2.79 | / | / | / |
| Heun | 15 | 30x2 | 0 | 2.42 | / | / | / |
| Adam2 | 16 | 16x2 | 0 | 2.42 | 237 | 0.80 | 0.60 |
| Adam4 | 16 | 16x2 | 0 | 2.27 | 243 | 0.80 | 0.60 |
| FlowTurbo | 6 | (7+3)x2 | 30408704(29M) | 3.93 | 223.6 | 0.79 | 0.56 |
| FlowTurbo | 8 | (8+2)x2 | 30408704(29M) | 3.63 | / | / | / |
| FlowTurbo | 10 | (12+2)x2 | 30408704(29M) | 2.69 | / | / | / |
| FlowTurbo | 15 | (17+3)x2 | 30408704(29M) | 2.22 | 248 | 0.81 | 0.60 |
| ours | 6 | 6x2 | 21 | 3.57 | 214 | 0.77 | 0.58 |
| ours | 7 | 7x2 | 28 | 2.78 | 229 | 0.79 | 0.60 |
| ours | 8 | 8x2 | 36 | 2.65 | 234 | 0.79 | 0.60 |
| ours | 10 | 10x2 | 55 | 2.40 | 238 | 0.79 | 0.60 |
| ours | 15 | 15x2 | 55 | 2.24 | 244 | 0.80 | 0.60 |

Q.4 ABLATION ON SEARCH SAMPLES

We ablate the number of search samples on the 10-step and 8-step solver settings. *Samples* means the total training samples the searched solver has seen. *Unique Samples* means the total distinct samples the searched solver has seen. Our searched solver converges fast and gets saturated near 30000 samples.

| iters(10-step-solver) | samples | unique samples | FID | IS | PR | Recall |
|---|---|---|---|---|---|---|
| 313 | 10000 | 10000 | 2.54 | 239 | 0.79 | 0.59 |
| 626 | 20000 | 10000 | 2.38 | 239 | 0.79 | 0.60 |
| 939 | 30000 | 10000 | 2.49 | 240 | 0.79 | 0.59 |
| 1252 | 40000 | 10000 | 2.29 | 239 | 0.80 | 0.60 |
| 1565 | 50000 | 10000 | 2.41 | 240 | 0.80 | 0.59 |
| 626 | 20000 | 20000 | 2.47 | 237 | 0.78 | 0.60 |
| 939 | 30000 | 30000 | 2.40 | 238 | 0.79 | 0.60 |
| 1252 | 40000 | 40000 | 2.48 | 237 | 0.80 | 0.59 |
| 1565 | 50000 | 50000 | 2.41 | 239 | 0.80 | 0.59 |

| iters(8-step-solver) | samples | unique samples | FID | IS | PR | Recall |
|---|---|---|---|---|---|---|
| 313 | 10000 | 10000 | 2.99 | 228 | 0.78 | 0.59 |
| 626 | 20000 | 10000 | 2.78 | 229 | 0.79 | 0.60 |
| 939 | 30000 | 10000 | 2.72 | 235 | 0.79 | 0.60 |
| 1252 | 40000 | 10000 | 2.67 | 228 | 0.79 | 0.60 |
| 1565 | 50000 | 10000 | 2.69 | 235 | 0.79 | 0.59 |
| 626 | 20000 | 20000 | 2.70 | 231 | 0.79 | 0.59 |
| 939 | 30000 | 30000 | 2.82 | 232 | 0.79 | 0.59 |
| 1252 | 40000 | 40000 | 2.79 | 231 | 0.79 | 0.60 |
| 1565 | 50000 | 50000 | 2.65 | 234 | 0.79 | 0.60 |

## Q.5 STOPPED EVALUATION AT 5 STEPS.

Since DM-nonuniform introduced the most effective online optimization solver before our search-based approach, we leveraged their results for comparison on DDPM models. We followed the evaluation pipeline established by DM-nonuniform to report performance within 5 and 10 optimization steps. In general, solver-based methods tend to exhibit inferior results under extremely limited numbers of function evaluations (NFE), such as 5 or 6 steps. As the solving difficulty increases and the number of searchable parameters decreases (e.g., only 10 searchable parameters for 4 steps and 6 searchable parameters for 3 steps), the performance of solver-based methods falls significantly behind that of distillation methods when limited to fewer than 5 steps. Notably, it is unlikely for solver-based methods to achieve performance comparable to or exceeding that of distillation methods, such as CM, given that their number of learnable parameters is tens of thousands of times larger than our searchable parameters.

Furthermore, integrating denoiser distillation with solver search holds significant promise for achieving even greater performance enhancements.

| | Steps | NFE-CFG | Extra-Paramters | FID | IS | PR | Recall |
|---|---|---|---|---|---|---|---|
| Euler | 1 | 1x2 | / | 300 | 2.32 | / | / |
| Euler | 50 | 50x2 | / | 2.23 | 244 | 0.80 | 0.59 |
| Adam2 | 3 | 3x2 | / | 41.2 | 68.6 | 0.44 | 0.46 |
| Adam2 | 4 | 4x2 | / | 15.25 | 133.6 | 0.65 | 0.50 |
| Adam2 | 5 | 5x2 | / | 8.96 | 170 | 0.73 | 0.53 |
| Adam2 | 6 | 6x2 | / | 6.35 | 191 | 0.76 | 0.55 |
| Adam2 | 15 | 15x2 | / | 2.49 | 236 | 0.79 | 0.59 |
| Adam4 | 15 | 15x2 | / | 2.33 | 242 | 0.80 | 0.59 |
| ours | 1 | 1x2 | 0 | 300 | 2.32 | / | / |
| ours | 3 | 3x2 | 6 | 39.3 | 68.6 | 0.46 | 0.52 |
| ours | 4 | 4x2 | 10 | 13.9 | 135 | 0.65 | 0.55 |
| ours | 5 | 5x2 | 15 | 4.52 | 194 | 0.75 | 0.58 |
| ours | 6 | 6x2 | 21 | 3.57 | 214 | 0.77 | 0.58 |
| ours | 15 | 15x2 | 55 | 2.24 | 244 | 0.80 | 0.60 |

## Q.6 ERROR BOUND ANALYSIS IN SECTION 4.3

Our primary objective is to design a compact search space that enables the identification of a solver that achieves near-optimal performance. To accomplish this, we must first establish the constituent components of the search space for the optimal solution. Notably, if the error bound is independent of the number of steps, our search can be limited to the coefficients alone. In fact, it can be proved that the error bound is dictated by the time selection and the coefficients.

## Q.7 REPHRASE NARRATIVE STYLE WRITING AS THEOREMS.

Thanks for your suggestions. We will re-organize the structure of our paper. We will add some summarization theorems in each subsection.

## Q.8 WHAT IS $\eta$ IN SECTION 4.3?

$\eta$ is a constant scalar. We will add more explanation of notations in the finial version.

## Q.9 RICHARDSON'S EXTRAPOLATION FOR SOLVING ODE

Yes, the Adams-like linear multi-step method employs Lagrange interpolation to determine its coefficients, which makes it feasible to substitute Lagrange interpolation with alternative interpolation (or extrapolation) techniquesFekete & Lóczi (2022), such as Richardson's method. Nevertheless, Richardson functions also solely rely on the variable $t$, without considering $x$.

## Q.10 SOLVER ACROSS DIFFERENT VARIANCE SCHEDULES

Since our solvers are searched on a specific noise scheduler and its corresponding pre-trained models, applying the searched coefficients and timesteps to other noise schedulers yields meaningless results. We have tried applied searched solver on SiT(Rectified flow) and DiT(DDPM with $\beta_{min} = 0.1, \beta_{max} = 20$) to SD1.5(DDPM with $\beta_{min} = 0.085, \beta_{max} = 12$), but the results were inconclusive. Notably, despite sharing the DDPM name, DiT and SD1.5 employ distinct $\beta_{min}, \beta_{max}$ values, thereby featuring different noise schedulers. A more in-depth discussion of these experiments can be found in Section(Extend to DDPM/VP).

## Q.11 SOLVER FOR DIFFERENT VARIANCE SCHEDULES

As every DDPM has a corresponding continuous VP scheduler, so we can transform the discreet DDPM into continuous VP, thus we successfully searched better solver compared to DPM-Solvers. The details can be found in Section 6. To put it simply, under the empowerment of our high-order solver, the performance of DDPM and FM does not differ significantly (8, 9, 10 steps), which contradicts the common belief that FM is stronger at limited sampling steps.

## Q.12 TEXT TO IMAGE METRICS RESULT

We take PixArt-alpha as the text-to-image model. We follow the evaluation pipeline of ADM and take COCO17-Val as the reference batch. We generate 5k images using DPM-Solver++, UniPC and our solver searched on DiT-XL/2-R256.

## Q.13 LIMITATIONS.

We place the limitation at the appendix, in order to provide more discussion space and obtain more insights from reviews. We copy the original limitation content and add more.

**Misalignd Reconstrucion loss and Performance.** Our proposed methods are specifically designed to minimize integral error within a limited number of steps. However, ablation studies reveal a mismatch between FID performance and Reconstruction error. To address this issue, we plan to enhance our searched solver by incorporating distribution matching supervision, thereby better aligning sampling quality.

| | Steps | FID | sFID | IS | PR | Recall |
|---|---|---|---|---|---|---|
| DPM++ | 5 | 60.0 | 209 | 25.59 | 0.36 | 0.20 |
| DPM++ | 8 | 38.4 | 116.9 | 33.0 | 0.50 | 0.36 |
| DPM++ | 10 | 35.6 | 114.7 | 33.7 | 0.53 | 0.37 |
| UniPC | 5 | 57.9 | 206.4 | 25.88 | 0.38 | 0.20 |
| UniPC | 8 | 37.6 | 115.3 | 33.3 | 0.51 | 0.36 |
| UniPC | 10 | 35.3 | 113.3 | 33.6 | 0.54 | 0.36 |
| Ours | 5 | 46.4 | 204 | 28.0 | 0.46 | 0.23 |
| Ours | 8 | 33.6 | 115.2 | 32.6 | 0.54 | 0.39 |
| Ours | 10 | 33.4 | 114.7 | 32.5 | 0.55 | 0.39 |

**Larger CFG Inference.** In the main paper, we demonstrate text-to-image visualization with a small CFG value. However, it is intuitive that utilizing a larger CFG would result in superior image quality. We attribute the inferior performance of large CFGs on our solver to the limitations of current naive solver structures and searching techniques. We hypothesize that incorporating predictor-corrector solver structures would enhance numerical stability and yield better images. Additionally, training with CFGs may also be beneficial.

**Resource Consumption** We can hard code the searched coefficients and timesteps into the program files. However, Compared to hand-crafted solvers, our solver still needs a searching process.

# A    PROOF OF PRE-INTEGRAL ERROR EXPECTATION

**Theorem A.1.** *Given sampling time interval $[t_i, t_{i+1}]$ and suppose $\mathcal{C}_j(\boldsymbol{x}) = g_j(\boldsymbol{x}) + b_i^j$, Adams-like linear multi-step methods will introduce an upper error bound of $(t_{i+1} - t_i)\mathbb{E}_{\boldsymbol{x}_i}||\sum_{j=0}^{i} \boldsymbol{v}_j g_j(\boldsymbol{x}_i)||$.*

*Our solver search(replacing $\mathcal{C}_j(\boldsymbol{x})$ with $\mathbb{E}_{\boldsymbol{x}_i}[\mathcal{C}_j(\boldsymbol{x}_i)]$) owns an upper error bound of $(t_{i+1} - t_i)\mathbb{E}_{\boldsymbol{x}_i}||\sum_{j=0}^{i} \boldsymbol{v}_j[g_j(\boldsymbol{x}_i) - \mathbb{E}_{\boldsymbol{x}_i} g_j(\boldsymbol{x}_i)]||$*

*Proof.* Suppose $\mathcal{C}_j(\boldsymbol{x}_i) = g_j(\boldsymbol{x}_i) + b_i^j$. Adams-like linear multi-step methods would not consider $x$-related interpolation. thus pre-integral coefficients of Adams-like linear multi-step methods will only reduce into $b$.

We obtain the error expectation of the pre-integral of Adams-like linear multi-step methods:

$$\mathbb{E}_{\boldsymbol{x}_i}||\sum_{j=0}^{i} \boldsymbol{v}_j[\mathcal{C}_j(\boldsymbol{x}_i)](t_{i+1} - t_i) - \sum_{j=0}^{i} \boldsymbol{v}_j b_i^j (t_{i+1} - t_i)|| \tag{21}$$

$$=\mathbb{E}_{\boldsymbol{x}_i}||\sum_{j=0}^{i} \boldsymbol{v}_j(t_{i+1} - t_i)[\mathcal{C}_j(\boldsymbol{x}_i) - b_i^j|| \tag{22}$$

$$=(t_{i+1} - t_i)\mathbb{E}_{\boldsymbol{x}_i}||\sum_{j=0}^{i} \boldsymbol{v}_j g_j(\boldsymbol{x}_i)|| \tag{23}$$

We obtain the error expectation of the pre-integral of our solver search methods:

$$\mathbb{E}_{\boldsymbol{x}_i}||\sum_{j=0}^{i} \boldsymbol{v}_j[\mathcal{C}_j(\boldsymbol{x}_i)](t_{i+1} - t_i) - \sum_{j=0}^{i} \boldsymbol{v}_j \mathbb{E}_{\boldsymbol{x}_i}[\mathcal{C}_j(\boldsymbol{x}_i)](t_{i+1} - t_i)|| \tag{24}$$

$$=\mathbb{E}_{\boldsymbol{x}_i}||\sum_{j=0}^{i} \boldsymbol{v}_j(t_{i+1} - t_i)[\mathcal{C}_j(\boldsymbol{x}_i) - \mathbb{E}_{\boldsymbol{x}_i}\mathcal{C}_j(\boldsymbol{x}_i)]|| \tag{25}$$

$$=(t_{i+1} - t_i)\mathbb{E}_{\boldsymbol{x}_i}||\sum_{j=0}^{i} \boldsymbol{v}_j[g_j(\boldsymbol{x}_i) - \mathbb{E}_{\boldsymbol{x}_i} g_j(\boldsymbol{x}_i)]|| \tag{26}$$

Next, define the optimization problem:

$$E = \mathbb{E}_{\boldsymbol{x}_i}||\sum_{j=0}^{i} \boldsymbol{v}_j[g_j(\boldsymbol{x}_i) - a_j]||_2^2.$$

We suppose different $v_j$ are orthogonal and $||v_j||_2^2 = 1$. As we leave $c_j^i$ as the expectation of $\mathcal{C}_j(\boldsymbol{x}_i)$, we will demonstrate this choice is optimal.

$$\frac{\partial E}{\partial a_j} = -2\mathbb{E}_{\boldsymbol{x}_i}(||v_j||_2^2(g_j(x_i) - a_j)) \tag{27}$$

Let $\frac{\partial E}{\partial a_j} = 0$, we obtain: $a_j = \frac{\mathbb{E}_{\boldsymbol{x}_i} g_i(x_i)||v_j||_2^2}{\mathbb{E}_{\boldsymbol{x}_i}||v_j||_2^2} = \mathbb{E}_{\boldsymbol{x}_i} g_j(\boldsymbol{x}_i) = \mathbb{E}_{\boldsymbol{x}_i}\mathcal{C}_j(\boldsymbol{x}_i) - b_i^j$.

So our searched solver has a lower and optimal error expectation:

$$(t_{i+1} - t_i)\mathbb{E}_{\boldsymbol{x}_i}||\sum_{j=0}^{i} \boldsymbol{v}_j[g_j(\boldsymbol{x}_i) - \mathbb{E}_{\boldsymbol{x}_i} g_j(\boldsymbol{x}_i)]|| \leq (t_{i+1} - t_i)\mathbb{E}_{\boldsymbol{x}_i}||\sum_{j=0}^{i} \boldsymbol{v}_j g_j(\boldsymbol{x}_i)|| \tag{28}$$

Recall Assumption 4.1, the integral upper error bound of universal interpolation $\mathcal{P}$ will be:

$$|| \int_{t_i}^{t_{i+1}} v(\boldsymbol{x}_t, t)dt - \sum_{j=0}^{i} \boldsymbol{v}_j \int_{t_i}^{t_{i+1}} \mathcal{P}(\boldsymbol{x}_t, t, \boldsymbol{x}_j, t_j)dt||. \tag{29}$$

$$= || \int_{t_i}^{t_{i+1}} v(\boldsymbol{x}_t, t)dt - \int_{t_i}^{t_{i+1}} \sum_{j=0}^{i} \mathcal{P}(\boldsymbol{x}_t, t, \boldsymbol{x}_j, t_j)\boldsymbol{v}_j dt||. \tag{30}$$

$$= || \int_{t_i}^{t_{i+1}} [v(\boldsymbol{x}_t, t) - \sum_{j=0}^{i} \mathcal{P}(\boldsymbol{x}_t, t, \boldsymbol{x}_j, t_j)\boldsymbol{v}_j]dt||. \tag{31}$$

$$< \int_{t_i}^{t_{i+1}} ||v(\boldsymbol{x}_t, t) - \sum_{j=0}^{i} \mathcal{P}(\boldsymbol{x}_t, t, \boldsymbol{x}_j, t_j)\boldsymbol{v}_j||dt. \tag{32}$$

$$< (t_{i+1} - t_i)[\mathcal{O}(d\boldsymbol{x}^m) + \mathcal{O}(dt^n)] \tag{33}$$

Combining Equation (33) and the error expectation of the pre-integral part, we will get the total error bound of the solver search.

$$|| \int_{t_i}^{t_{i+1}} v(\boldsymbol{x}_t, t)dt - \sum_{j=0}^{i} \boldsymbol{v}_j \mathbb{E}_{\boldsymbol{x}_i}[\mathcal{C}_j(\boldsymbol{x}_i)](t_{i+1} - t_i)||. \tag{34}$$

$$= || \int_{t_i}^{t_{i+1}} v(\boldsymbol{x}_t, t)dt - \sum_{j=0}^{i} \boldsymbol{v}_j \int_{t_i}^{t_{i+1}} \mathcal{P}(\boldsymbol{x}_t, t, \boldsymbol{x}_j, t_j)dt+ \tag{35}$$

$$\sum_{j=0}^{i} \boldsymbol{v}_j \int_{t_i}^{t_{i+1}} \mathcal{P}(\boldsymbol{x}_t, t, \boldsymbol{x}_j, t_j)dt - \sum_{j=0}^{i} \boldsymbol{v}_j \mathbb{E}_{\boldsymbol{x}_i}[\mathcal{C}_j(\boldsymbol{x}_i)](t_{i+1} - t_i)||. \tag{36}$$

$$< || \int_{t_i}^{t_{i+1}} v(\boldsymbol{x}_t, t)dt - \sum_{j=0}^{i} \boldsymbol{v}_j \int_{t_i}^{t_{i+1}} \mathcal{P}(\boldsymbol{x}_t, t, \boldsymbol{x}_j, t_j)dt||+ \tag{37}$$

$$|| \sum_{j=0}^{i} \boldsymbol{v}_j \int_{t_i}^{t_{i+1}} \mathcal{P}(\boldsymbol{x}_t, t, \boldsymbol{x}_j, t_j)dt - \sum_{j=0}^{i} \boldsymbol{v}_j \mathbb{E}_{\boldsymbol{x}_i}[\mathcal{C}_j(\boldsymbol{x}_i)](t_{i+1} - t_i)||. \tag{38}$$

$$= || \int_{t_i}^{t_{i+1}} v(\boldsymbol{x}_t, t)dt - \sum_{j=0}^{i} \boldsymbol{v}_j \int_{t_i}^{t_{i+1}} \mathcal{P}(\boldsymbol{x}_t, t, \boldsymbol{x}_j, t_j)dt||+ \tag{39}$$

$$|| \sum_{j=0}^{i} \boldsymbol{v}_j[\mathcal{C}_j(\boldsymbol{x}_i)](t_{i+1} - t_i) - \sum_{j=0}^{i} \boldsymbol{v}_j \mathbb{E}_{\boldsymbol{x}_i}[\mathcal{C}_j(\boldsymbol{x}_i)](t_{i+1} - t_i)||. \tag{40}$$

$$< (t_{i+1} - t_i)[\mathcal{O}(d\boldsymbol{x}^m) + \mathcal{O}(dt^n)] + (t_{i+1} - t_i)\mathbb{E}_{\boldsymbol{x}_i}|| \sum_{j=0}^{i} \boldsymbol{v}_j[g_j(\boldsymbol{x}_i) - \mathbb{E}_{\boldsymbol{x}_i} g_j(\boldsymbol{x}_i)]|| \tag{41}$$

$$< (t_{i+1} - t_i)([\mathcal{O}(d\boldsymbol{x}^m) + \mathcal{O}(dt^n)] + \mathbb{E}_{\boldsymbol{x}_i}|| \sum_{j=0}^{i} \boldsymbol{v}_j[g_j(\boldsymbol{x}_i) - \mathbb{E}_{\boldsymbol{x}_i} g_j(\boldsymbol{x}_i)]||) \tag{42}$$

Since $((\mathcal{O}(d\boldsymbol{x}^m) + \mathcal{O}(dt^n))$ is much smaller than $\mathbb{E}_{\boldsymbol{x}_i}|| \sum_{j=0}^{i} \boldsymbol{v}_j[g_j(\boldsymbol{x}_i) - \mathbb{E}_{\boldsymbol{x}_i} g_j(\boldsymbol{x}_i)]||$. We can omit the $((\mathcal{O}(d\boldsymbol{x}^m) + \mathcal{O}(dt^n))$ term.

$\square$

## B  PROOF OF TOTAL UPPER ERROR BOUND

**Theorem B.1.** *Compared to Adams-like linear multi-step methods. Our Solver search has a small upper error bound.*

*The total upper error bound of Adams-like linear multi-step methods is:*

$$\sum_{i=0}^{N-1}(\frac{1}{N})\sum_{j=0}^{i}\eta|b_i^j| + \mathbb{E}_{\boldsymbol{x}_i}||\sum_{j=0}^{i}\boldsymbol{v}_j[g_j(\boldsymbol{x}_i)]||)$$

*The total upper error bound of Our solver search is:*

$$\sum_{i=0}^{N-1}(t_{i+1}-t_i)\sum_{j=0}^{i}\eta|\mathbb{E}_{\boldsymbol{x}_i}g_j(\boldsymbol{x}_i)+b_i^j| + \mathbb{E}_{\boldsymbol{x}_i}||\sum_{j=0}^{i}\boldsymbol{v}_jg_j(\boldsymbol{x}_i)-\mathbb{E}_{\boldsymbol{x}_i}g_j(\boldsymbol{x}_i)||)$$

*Proof.* We donate the continuous integral result of the ideal velocity field $\hat{\boldsymbol{v}}$ as $\hat{\boldsymbol{x}}$, the solved integral result of the ideal velocity field $\hat{\boldsymbol{v}}$ as $\hat{\boldsymbol{x}}_N$, the continuous integral result of the pre-trained velocity model $\boldsymbol{v}_\theta$ as $\hat{\boldsymbol{x}}$, the solved integral result of the pre-trained velocity model $\boldsymbol{v}_\theta$ as $\boldsymbol{x}_N$.

$$\boldsymbol{x}_N = \boldsymbol{\epsilon} + \sum_{i=0}^{N-1}\sum_{j=0}^{i}\boldsymbol{v}_jc_i^j(t_{i+1}-t_i) \tag{43}$$

The error caused by the non-ideal velocity estimation model can be formulated in the following equation. we can employ triangular inequalities to obtain the error-bound $||\boldsymbol{x}_N-\hat{\boldsymbol{x}}_N||$, which is related to solver coefficients and timestep choices.

$$||\boldsymbol{x}_N-\hat{\boldsymbol{x}}_N|| = |\sum_{i=0}^{N-1}\sum_{j=0}^{i}(\boldsymbol{v}_j-\hat{\boldsymbol{v}}_j)c_i^j(t_{i+1}-t_i)|$$

$$\leq \sum_{i=0}^{N-1}\sum_{j=0}^{i}|(\boldsymbol{v}_j-\hat{\boldsymbol{v}}_j)c_i^j(t_{i+1}-t_i)|$$

$$\leq \sum_{i=0}^{N-1}\sum_{j=0}^{i}|\boldsymbol{v}_j-\hat{\boldsymbol{v}}_j)| \times |c_i^j(t_{i+1}-t_i)|$$

$$\leq \eta\sum_{i=0}^{N-1}\sum_{j=0}^{i}|c_i^j(t_{i+1}-t_i)|$$

The total error of our searched solver is:

$$||\boldsymbol{x}_N-\hat{\boldsymbol{x}}||$$
$$=||\boldsymbol{x}_N-\hat{\boldsymbol{x}}_N+\hat{\boldsymbol{x}}_N-\hat{\boldsymbol{x}}||$$
$$\leq||\boldsymbol{x}_N-\hat{\boldsymbol{x}}_N||+||\hat{\boldsymbol{x}}_N-\hat{\boldsymbol{x}}||$$
$$\leq\eta\sum_{i=0}^{N-1}\sum_{j=0}^{i}|c_i^j(t_{i+1}-t_i)|+$$
$$\sum_{i=0}^{N-1}(t_{i+1}-t_i)(\mathcal{O}(d\boldsymbol{x}^m)+\mathcal{O}(dt^n)+\mathbb{E}_{\boldsymbol{x}_i}||\sum_{j=0}^{i}\boldsymbol{v}_j[g_j(\boldsymbol{x}_i)-\mathbb{E}_{\boldsymbol{x}_i}g_j(\boldsymbol{x}_i)]||)$$
$$\approx\sum_{i=0}^{N-1}\eta\sum_{j=0}^{i}|c_i^j(t_{i+1}-t_i)|+(t_{i+1}-t_i)\mathbb{E}_{\boldsymbol{x}_i}||\sum_{j=0}^{i}\boldsymbol{v}_j[g_j(\boldsymbol{x}_i)-\mathbb{E}_{\boldsymbol{x}_i}g_j(\boldsymbol{x}_i)]||)$$
$$=\sum_{i=0}^{N-1}(t_{i+1}-t_i)\sum_{j=0}^{i}\eta|\mathbb{E}_{\boldsymbol{x}_i}g_j(\boldsymbol{x}_i)+b_i^j| + \mathbb{E}_{\boldsymbol{x}_i}||\sum_{j=0}^{i}\boldsymbol{v}_j[g_j(\boldsymbol{x}_i)-\mathbb{E}_{\boldsymbol{x}_i}g_j(\boldsymbol{x}_i)]||)$$

The total error of Adams-like linear multi-step method is:

$$\sum_{i=0}^{N-1}(\frac{1}{N})\sum_{j=0}^{i}\eta|b_i^j| + \mathbb{E}_{\boldsymbol{x}_i}||\sum_{j=0}^{i}\boldsymbol{v}_j[g_j(\boldsymbol{x}_i)]||)$$

Obviously, as $(\sum_{j=0}^{i} \eta |b_i^j| + \mathbb{E}_{\boldsymbol{x}_i} || \sum_{j=0}^{i} \boldsymbol{v}_j [g_j(\boldsymbol{x}_i)] ||)$ is not equal between different timestep intervals, Optimized timesteps owns smaller upper error bound than uniform timesteps.

Recall that $\eta \ll ||v_j||$, the error is mainly determined by $\mathbb{E}_{\boldsymbol{x}_i} || \sum_{j=0}^{i} \boldsymbol{v}_j [g_j(\boldsymbol{x}_i)] ||$.

Recall that $\mathbb{E}_{\boldsymbol{x}_i} || \sum_{j=0}^{i} \boldsymbol{v}_j [g_j(\boldsymbol{x}_i) - \mathbb{E}_{\boldsymbol{x}_i} g_j(\boldsymbol{x}_i)] || \leq \mathbb{E}_{\boldsymbol{x}_i} || \sum_{j=0}^{i} \boldsymbol{v}_j [g_j(\boldsymbol{x}_i)] ||$, thus our solver search has a minimal upper error bound because we search coefficients and timesteps simultaneously.

$\square$

## C  SEARCHED PARAMETERS

We provide the searched parameters $\Delta t$ and $c_i^j$. Note $c_i^j$ needs to be converted into $\mathcal{M}$ follwing Algorithm 1.

### C.1  SOLVER SEARCHED ON SIT-XL/2

| NFE | TimeDeltas $\Delta t$ | Coeffcients $c_i^j$ |
|---|---|---|
| 5 | $\begin{bmatrix} 0.0424 \\ 0.1225 \\ 0.2144 \\ 0.3073 \\ 0.3135 \end{bmatrix}$ | $\begin{bmatrix} 0.0 & 0.0 & 0.0 & 0.0 & 0.0 \\ -1.17 & 0.0 & 0.0 & 0.0 & 0.0 \\ 1.07 & -1.83 & 0.0 & 0.0 & 0.0 \\ 0.0 & 0.0 & -0.93 & 0.0 & 0.0 \\ 0.0 & 0.0 & 0.0 & -0.71 & 0.0 \end{bmatrix}$ |
| 6 | $\begin{bmatrix} 0.0389 \\ 0.0976 \\ 0.161 \\ 0.2046 \\ 0.2762 \\ 0.2217 \end{bmatrix}$ | $\begin{bmatrix} 0.0 & 0.0 & 0.0 & 0.0 & 0.0 & 0.0 \\ -1.04 & 0.0 & 0.0 & 0.0 & 0.0 & 0.0 \\ 1.62 & -2.98 & 0.0 & 0.0 & 0.0 & 0.0 \\ -1.32 & 2.52 & -2.04 & 0.0 & 0.0 & 0.0 \\ 0.0 & 0.0 & 0.0 & -0.76 & 0.0 & 0.0 \\ 0.0 & 0.0 & 0.0 & 0.0 & -0.66 & 0.0 \end{bmatrix}$ |
| 7 | $\begin{bmatrix} 0.0299 \\ 0.0735 \\ 0.1119 \\ 0.1451 \\ 0.1959 \\ 0.2698 \\ 0.1738 \end{bmatrix}$ | $\begin{bmatrix} 0.0 & 0.0 & 0.0 & 0.0 & 0.0 & 0.0 & 0.0 \\ -0.93 & 0.0 & 0.0 & 0.0 & 0.0 & 0.0 & 0.0 \\ 1.23 & -2.31 & 0.0 & 0.0 & 0.0 & 0.0 & 0.0 \\ -0.59 & 1.53 & -2.09 & 0.0 & 0.0 & 0.0 & 0.0 \\ -0.09 & -0.07 & 0.99 & -1.91 & 0.0 & 0.0 & 0.0 \\ 0.05 & -0.21 & 0.09 & 0.55 & -1.47 & 0.0 & 0.0 \\ -0.05 & 0.19 & -0.31 & 0.37 & 0.67 & -1.79 & 0.0 \end{bmatrix}$ |
| 8 | $\begin{bmatrix} 0.0303 \\ 0.0702 \\ 0.0716 \\ 0.1112 \\ 0.1501 \\ 0.1833 \\ 0.2475 \\ 0.1358 \end{bmatrix}$ | $\begin{bmatrix} 0.0 & 0.0 & 0.0 & 0.0 & 0.0 & 0.0 & 0.0 & 0.0 \\ -0.92 & 0.0 & 0.0 & 0.0 & 0.0 & 0.0 & 0.0 & 0.0 \\ 0.78 & -1.7 & 0.0 & 0.0 & 0.0 & 0.0 & 0.0 & 0.0 \\ 0.06 & 0.52 & -1.76 & 0.0 & 0.0 & 0.0 & 0.0 & 0.0 \\ -0.02 & -0.16 & 0.98 & -1.8 & 0.0 & 0.0 & 0.0 & 0.0 \\ -0.02 & -0.12 & 0.22 & 0.24 & -1.36 & 0.0 & 0.0 & 0.0 \\ -0.1 & 0.06 & -0.02 & 0.18 & 0.12 & -1.1 & 0.0 & 0.0 \\ -0.16 & 0.14 & -0.02 & -0.02 & 0.38 & 0.32 & -1.72 & 0.0 \end{bmatrix}$ |
| 9 | $\begin{bmatrix} 0.028 \\ 0.0624 \\ 0.0717 \\ 0.0894 \\ 0.1092 \\ 0.1307 \\ 0.1729 \\ 0.2198 \\ 0.1159 \end{bmatrix}$ | $\begin{bmatrix} 0.0 & 0.0 & 0.0 & 0.0 & 0.0 & 0.0 & 0.0 & 0.0 & 0.0 \\ -0.93 & 0.0 & 0.0 & 0.0 & 0.0 & 0.0 & 0.0 & 0.0 & 0.0 \\ 0.63 & -1.29 & 0.0 & 0.0 & 0.0 & 0.0 & 0.0 & 0.0 & 0.0 \\ 0.39 & -0.11 & -1.41 & 0.0 & 0.0 & 0.0 & 0.0 & 0.0 & 0.0 \\ -0.07 & -0.05 & 0.83 & -1.59 & 0.0 & 0.0 & 0.0 & 0.0 & 0.0 \\ 0.07 & -0.11 & 0.27 & 0.27 & -1.53 & 0.0 & 0.0 & 0.0 & 0.0 \\ -0.05 & 0.03 & 0.01 & 0.15 & 0.17 & -1.15 & 0.0 & 0.0 & 0.0 \\ -0.21 & 0.27 & -0.07 & -0.03 & 0.19 & 0.09 & -0.99 & 0.0 & 0.0 \\ -0.15 & 0.15 & 0.03 & -0.09 & 0.25 & 0.25 & 0.21 & -1.71 & 0.0 \end{bmatrix}$ |
| 10 | $\begin{bmatrix} 0.0279 \\ 0.0479 \\ 0.0646 \\ 0.0659 \\ 0.1045 \\ 0.1066 \\ 0.1355 \\ 0.1622 \\ 0.1942 \\ 0.0908 \end{bmatrix}$ | $\begin{bmatrix} 0.0 & 0.0 & 0.0 & 0.0 & 0.0 & 0.0 & 0.0 & 0.0 & 0.0 & 0.0 \\ -0.95 & 0.0 & 0.0 & 0.0 & 0.0 & 0.0 & 0.0 & 0.0 & 0.0 & 0.0 \\ 0.59 & -1.17 & 0.0 & 0.0 & 0.0 & 0.0 & 0.0 & 0.0 & 0.0 & 0.0 \\ 0.35 & -0.11 & -1.45 & 0.0 & 0.0 & 0.0 & 0.0 & 0.0 & 0.0 & 0.0 \\ -0.13 & 0.01 & 0.75 & -1.49 & 0.0 & 0.0 & 0.0 & 0.0 & 0.0 & 0.0 \\ 0.05 & -0.05 & 0.31 & 0.29 & -1.59 & 0.0 & 0.0 & 0.0 & 0.0 & 0.0 \\ 0.05 & -0.03 & -0.09 & 0.23 & 0.17 & -1.19 & 0.0 & 0.0 & 0.0 & 0.0 \\ -0.03 & 0.07 & -0.09 & -0.03 & 0.27 & -0.03 & -0.91 & 0.0 & 0.0 & 0.0 \\ -0.15 & 0.17 & 0.03 & -0.09 & 0.05 & 0.09 & 0.05 & -0.79 & 0.0 & 0.0 \\ -0.17 & 0.11 & 0.15 & 0.03 & 0.05 & 0.25 & 0.05 & -0.07 & -1.49 & 0.0 \end{bmatrix}$ |

### C.2 SOLVER SEARCHED ON FLOWDCN-B/2

| NFE | TimeDeltas $\Delta t$ | Coeffcients $c_i^j$ |
|---|---|---|
| 5 | $\begin{bmatrix} 0.0521 \\ 0.1475 \\ 0.2114 \\ 0.2797 \\ 0.3092 \end{bmatrix}$ | $\begin{bmatrix} 0.0 & 0.0 & 0.0 & 0.0 & 0.0 \\ -1.26 & 0.0 & 0.0 & 0.0 & 0.0 \\ 1.38 & -2.26 & 0.0 & 0.0 & 0.0 \\ 0.0 & 0.0 & -0.92 & 0.0 & 0.0 \\ 0.0 & 0.0 & 0.0 & -0.7 & 0.0 \end{bmatrix}$ |
| 6 | $\begin{bmatrix} 0.0391 \\ 0.0924 \\ 0.165 \\ 0.2015 \\ 0.2511 \\ 0.2511 \end{bmatrix}$ | $\begin{bmatrix} 0.0 & 0.0 & 0.0 & 0.0 & 0.0 & 0.0 \\ -1.22 & 0.0 & 0.0 & 0.0 & 0.0 & 0.0 \\ 1.12 & -2.0 & 0.0 & 0.0 & 0.0 & 0.0 \\ -0.3 & 0.9 & -1.56 & 0.0 & 0.0 & 0.0 \\ 0.0 & 0.0 & 0.0 & -0.74 & 0.0 & 0.0 \\ 0.0 & 0.0 & 0.0 & 0.0 & -0.62 & 0.0 \end{bmatrix}$ |
| 7 | $\begin{bmatrix} 0.0387 \\ 0.0748 \\ 0.103 \\ 0.1537 \\ 0.184 \\ 0.234 \\ 0.2117 \end{bmatrix}$ | $\begin{bmatrix} 0.0 & 0.0 & 0.0 & 0.0 & 0.0 & 0.0 & 0.0 \\ -1.11 & 0.0 & 0.0 & 0.0 & 0.0 & 0.0 & 0.0 \\ 1.03 & -1.99 & 0.0 & 0.0 & 0.0 & 0.0 & 0.0 \\ 0.07 & 0.43 & -1.57 & 0.0 & 0.0 & 0.0 & 0.0 \\ -0.21 & -0.15 & 1.53 & -2.29 & 0.0 & 0.0 & 0.0 \\ -0.05 & 0.07 & -0.23 & 0.61 & -1.33 & 0.0 & 0.0 \\ -0.17 & 0.31 & -0.41 & 0.17 & 0.59 & -1.31 & 0.0 \end{bmatrix}$ |
| 8 | $\begin{bmatrix} 0.0071 \\ 0.0613 \\ 0.078 \\ 0.1163 \\ 0.1421 \\ 0.188 \\ 0.2077 \\ 0.1996 \end{bmatrix}$ | $\begin{bmatrix} 0.0 & 0.0 & 0.0 & 0.0 & 0.0 & 0.0 & 0.0 & 0.0 \\ -2.43 & 0.0 & 0.0 & 0.0 & 0.0 & 0.0 & 0.0 & 0.0 \\ 0.61 & -1.55 & 0.0 & 0.0 & 0.0 & 0.0 & 0.0 & 0.0 \\ 0.99 & -0.11 & -2.07 & 0.0 & 0.0 & 0.0 & 0.0 & 0.0 \\ 0.05 & -0.49 & 1.33 & -1.93 & 0.0 & 0.0 & 0.0 & 0.0 \\ 0.05 & -0.33 & 0.23 & 0.73 & -1.71 & 0.0 & 0.0 & 0.0 \\ -0.09 & 0.25 & -0.29 & 0.05 & 0.61 & -1.45 & 0.0 & 0.0 \\ -0.23 & 0.21 & -0.01 & -0.25 & 0.25 & 0.41 & -1.25 & 0.0 \end{bmatrix}$ |
| 9 | $\begin{bmatrix} 0.0017 \\ 0.051 \\ 0.0636 \\ 0.0911 \\ 0.1007 \\ 0.1443 \\ 0.1694 \\ 0.191 \\ 0.1872 \end{bmatrix}$ | $\begin{bmatrix} 0.0 & 0.0 & 0.0 & 0.0 & 0.0 & 0.0 & 0.0 & 0.0 & 0.0 \\ -6.19 & 0.0 & 0.0 & 0.0 & 0.0 & 0.0 & 0.0 & 0.0 & 0.0 \\ -0.11 & -0.81 & 0.0 & 0.0 & 0.0 & 0.0 & 0.0 & 0.0 & 0.0 \\ 0.73 & -0.17 & -1.37 & 0.0 & 0.0 & 0.0 & 0.0 & 0.0 & 0.0 \\ 0.31 & -0.05 & 0.19 & -1.45 & 0.0 & 0.0 & 0.0 & 0.0 & 0.0 \\ 0.03 & -0.23 & 0.29 & 0.35 & -1.35 & 0.0 & 0.0 & 0.0 & 0.0 \\ -0.19 & 0.05 & 0.01 & 0.21 & 0.25 & -1.23 & 0.0 & 0.30 & 0.0 \\ -0.23 & 0.21 & -0.13 & 0.17 & 0.09 & 0.09 & -1.09 & 0.0 & 0.0 \\ -0.17 & 0.15 & 0.11 & -0.19 & 0.03 & 0.23 & 0.17 & -1.21 & 0.0 \end{bmatrix}$ |
| 10 | $\begin{bmatrix} 0.0016 \\ 0.0538 \\ 0.0347 \\ 0.0853 \\ 0.0853 \\ 0.1198 \\ 0.1351 \\ 0.165 \\ 0.1788 \\ 0.1406 \end{bmatrix}$ | $\begin{bmatrix} 0.0 & 0.0 & 0.0 & 0.0 & 0.0 & 0.0 & 0.0 & 0.0 & 0.0 & 0.0 \\ -7.8801 & 0.0 & 0.0 & 0.0 & 0.0 & 0.0 & 0.0 & 0.0 & 0.0 & 0.0 \\ -0.4 & -0.74 & 0.0 & 0.0 & 0.0 & 0.0 & 0.0 & 0.0 & 0.0 & 0.0 \\ 0.48 & -0.18 & -0.86 & 0.0 & 0.0 & 0.0 & 0.0 & 0.0 & 0.0 & 0.0 \\ 0.26 & -0.04 & -0.04 & -1.28 & 0.0 & 0.0 & 0.0 & 0.0 & 0.0 & 0.0 \\ 0.0 & -0.06 & 0.26 & 0.26 & -1.42 & 0.0 & 0.0 & 0.0 & 0.0 & 0.0 \\ -0.1 & -0.06 & 0.08 & 0.2 & 0.22 & -1.24 & 0.0 & 0.0 & 0.0 & 0.0 \\ -0.18 & 0.14 & -0.08 & 0.1 & 0.08 & 0.14 & -1.06 & 0.0 & 0.0 & 0.0 \\ -0.12 & 0.16 & -0.1 & 0.04 & 0.08 & 0.06 & 0.08 & -1.02 & 0.0 & 0.0 \\ -0.16 & 0.02 & 0.14 & 0.0 & -0.14 & 0.08 & 0.14 & 0.34 & -1.38 & 0.0 \end{bmatrix}$ |

### C.3 SOLVER SEARCHED ON DiT-XL/2

| NFE | TimeDeltas $\Delta t$ | Coeffcients $c_i^j$ |
|---|---|---|
| 5 | $\begin{bmatrix} 0.2582 \\ 0.1766 \\ 0.1766 \\ 0.2156 \\ 0.1731 \end{bmatrix}$ | $\begin{bmatrix} 0.0 & 0.0 & 0.0 & 0.0 & 0.0 \\ -1.43 & 0.0 & 0.0 & 0.0 & 0.0 \\ 0.93 & -1.55 & 0.0 & 0.0 & 0.0 \\ 0.0 & 0.0 & -0.69 & 0.0 & 0.0 \\ 0.0 & 0.0 & 0.0 & -0.59 & 0.0 \end{bmatrix}$ |
| 6 | $\begin{bmatrix} 0.2483 \\ 0.1506 \\ 0.1476 \\ 0.1568 \\ 0.1733 \\ 0.1233 \end{bmatrix}$ | $\begin{bmatrix} 0.0 & 0.0 & 0.0 & 0.0 & 0.0 & 0.0 \\ -1.36 & 0.0 & 0.0 & 0.0 & 0.0 & 0.0 \\ 0.9 & -1.84 & 0.0 & 0.0 & 0.0 & 0.0 \\ -0.08 & 0.5 & -1.08 & 0.0 & 0.0 & 0.0 \\ 0.0 & 0.0 & 0.0 & -0.56 & 0.0 & 0.0 \\ 0.0 & 0.0 & 0.0 & 0.0 & -0.56 & 0.0 \end{bmatrix}$ |
| 7 | $\begin{bmatrix} 0.2241 \\ 0.1415 \\ 0.1205 \\ 0.1158 \\ 0.1443 \\ 0.1627 \\ 0.0911 \end{bmatrix}$ | $\begin{bmatrix} 0.0 & 0.0 & 0.0 & 0.0 & 0.0 & 0.0 & 0.0 \\ -1.38 & 0.0 & 0.0 & 0.0 & 0.0 & 0.0 & 0.0 \\ 1.08 & -2.02 & 0.0 & 0.0 & 0.0 & 0.0 & 0.0 \\ -0.28 & 0.78 & -1.52 & 0.0 & 0.0 & 0.0 & 0.0 \\ -1.4901e-08 & -0.1 & 0.64 & -1.5 & 0.0 & 0.0 & 0.0 \\ 0.06 & -0.06 & -0.06 & 0.26 & -1.0 & 0.0 & 0.0 \\ 0.0 & -0.1 & 0.02 & 0.2 & 0.26 & -1.12 & 0.0 \end{bmatrix}$ |
| 8 | $\begin{bmatrix} 0.2033 \\ 0.1476 \\ 0.1094 \\ 0.099 \\ 0.1116 \\ 0.1233 \\ 0.131 \\ 0.0748 \end{bmatrix}$ | $\begin{bmatrix} 0.0 & 0.0 & 0.0 & 0.0 & 0.0 & 0.0 & 0.0 & 0.0 \\ -1.14 & 0.0 & 0.0 & 0.0 & 0.0 & 0.0 & 0.0 & 0.0 \\ 0.8 & -1.76 & 0.0 & 0.0 & 0.0 & 0.0 & 0.0 & 0.0 \\ 0.02 & 0.48 & -1.62 & 0.0 & 0.0 & 0.0 & 0.0 & 0.0 \\ -0.12 & 0.06 & 0.62 & -1.42 & 0.0 & 0.0 & 0.0 & 0.0 \\ 0.04 & -0.1 & 0.12 & 0.16 & -1.04 & 0.0 & 0.0 & 0.0 \\ 0.06 & -0.04 & -0.06 & 0.08 & -0.08 & -0.56 & 0.0 & 0.0 \\ -0.02 & -0.04 & -0.04 & 0.12 & 0.14 & 0.04 & -0.9 & 0.0 \end{bmatrix}$ |
| 9 | $\begin{bmatrix} 0.1959 \\ 0.1313 \\ 0.1142 \\ 0.0863 \\ 0.0898 \\ 0.0916 \\ 0.1119 \\ 0.1054 \\ 0.0735 \end{bmatrix}$ | $\begin{bmatrix} 0.0 & 0.0 & 0.0 & 0.0 & 0.0 & 0.0 & 0.0 & 0.0 & 0.0 \\ -1.28 & 0.0 & 0.0 & 0.0 & 0.0 & 0.0 & 0.0 & 0.0 & 0.0 \\ 0.78 & -1.62 & 0.0 & 0.0 & 0.0 & 0.0 & 0.0 & 0.0 & 0.0 \\ -0.02 & 0.44 & -1.48 & 0.0 & 0.0 & 0.0 & 0.0 & 0.0 & 0.0 \\ -0.1 & 0.16 & 0.36 & -1.3 & 0.0 & 0.0 & 0.0 & 0.0 & 0.0 \\ -0.06 & -0.04 & 0.22 & 0.12 & -1.08 & 0.0 & 0.0 & 0.0 & 0.0 \\ 0.08 & -0.1 & -0.04 & 0.24 & -0.06 & -0.86 & 0.0 & 0.0 & 0.0 \\ 0.04 & -0.04 & -0.04 & 0.0 & 0.06 & -0.08 & -0.5 & 0.0 & 0.0 \\ -0.04 & 0.0 & 0.0 & -0.02 & 0.14 & 0.02 & 0.0 & -0.74 & 0.0 \end{bmatrix}$ |
| 10 | $\begin{bmatrix} 0.2174 \\ 0.1123 \\ 0.1037 \\ 0.0724 \\ 0.0681 \\ 0.0816 \\ 0.0938 \\ 0.0977 \\ 0.0849 \\ 0.0681 \end{bmatrix}$ | $\begin{bmatrix} 0.0 & 0.0 & 0.0 & 0.0 & 0.0 & 0.0 & 0.0 & 0.0 & 0.0 & 0.0 \\ -1.17 & 0.0 & 0.0 & 0.0 & 0.0 & 0.0 & 0.0 & 0.0 & 0.0 & 0.0 \\ 0.35 & -0.99 & 0.0 & 0.0 & 0.0 & 0.0 & 0.0 & 0.0 & 0.0 & 0.0 \\ 0.25 & -0.11 & -0.99 & 0.0 & 0.0 & 0.0 & 0.0 & 0.0 & 0.0 & 0.0 \\ 0.03 & 0.05 & -0.07 & -0.85 & 0.0 & 0.0 & 0.0 & 0.0 & 0.0 & 0.0 \\ -0.03 & 0.03 & 0.25 & -0.09 & -0.93 & 0.0 & 0.0 & 0.0 & 0.0 & 0.0 \\ -0.01 & -0.03 & -0.01 & 0.21 & -0.11 & -0.67 & 0.0 & 0.0 & 0.0 & 0.0 \\ 0.01 & -0.03 & -0.03 & 0.07 & 0.09 & -0.03 & -0.81 & 0.0 & 0.0 & 0.0 \\ 0.03 & -0.03 & -0.03 & -0.03 & 0.05 & 0.01 & -0.11 & -0.27 & 0.0 & 0.0 \\ -0.01 & -0.01 & -0.01 & -0.01 & 0.03 & 0.07 & -0.01 & -0.05 & -0.57 & 0.0 \end{bmatrix}$ |

## D SOLVER CODE

### D.1 DDPM/VP CODE

```python
# corresponding to DDPM(beta_min=0.0001 beta_max=0.02)
class VPScheduler:
```

```python
    def __init__(
            self,
            beta_min=0.1,
            beta_max=20,
    ):
        super().__init__()
        self.beta_min = beta_min
        self.beta_d = beta_max - beta_min
    def beta(self, t) -> Tensor:
        t = torch.clamp(t, min=1e-3, max=1)
        return (self.beta_min + (self.beta_d * t)).view(-1, 1, 1, 1)

    def sigma(self, t) -> Tensor:
        t = torch.clamp(t, min=1e-3, max=1)
        inter_beta:Tensor = 0.5*self.beta_d*t**2 + self.beta_min* t
        return (1-torch.exp_(-inter_beta)).sqrt().view(-1, 1, 1, 1)

    def alpha(self, t) -> Tensor:
        t = torch.clamp(t, min=1e-3, max=1)
        inter_beta: Tensor = 0.5 * self.beta_d * t ** 2 + self.beta_min * t
        return torch.exp(-0.5*inter_beta).view(-1, 1, 1, 1)

class Scheduler(SchedulerMixin, ConfigMixin):
    @register_to_config
    def __init__(
        self,
        num_train_timesteps: int = 1000,
    ):
        self.num_train_timesteps = num_train_timesteps
        self.vp_scheduler = VPScheduler()
        self.init_noise_sigma = 1.0
        self.buffer = []
        self._index = 0
    def set_timesteps(self, num_inference_steps: int, device: torch.device):
        # index Params according to num_inference_steps
        self._timedeltas = ...
        self._coeffs = ...
        self._contiguous_timestep = [0.999,]
        for i in range(num_inference_steps-1):
            t = max(self._contiguous_timestep[-1] - self._timedeltas[i], 0.0)
            self._timestep.append(t)
        self.timesteps = torch.tensor(self._timestep)*self.num_train_timesteps
        self.timesteps = self.timesteps.to(torch.int64)
        self._contiguous_timestep = torch.tensor(self._contiguous_timestep)
        self.num_inference_steps = num_inference_steps

    def step(
        self,
        eps: torch.Tensor,
        timestep: int,
        x: torch.Tensor,
        return_dict: bool = True,
    ) -> Tuple:
        if timestep == self.num_train_timesteps -1:
            self.buffer.clear()
            self._index = 0
        t_cur = self._timestep[self._index]
        dt = self._timedeltas[self._index]
        sigma = self.vp_scheduler.sigma(t_cur)
```

```
1350        alpha = self.vp_scheduler.alpha(t_cur)
1351        lamda = (alpha / sigma)
1352        sigma_next = self.vp_scheduler.sigma(t_cur - dt)
1353        alpha_next = self.vp_scheduler.alpha(t_cur - dt)
1354        lamda_next = (alpha_next / sigma_next)
1355        x0 = (x - sigma * eps) / alpha
1356        self.buffer.append(x0)
1357        dpmx = torch.zeros_like(x0)
1358        sum_solver_coeff = 0.0
1359        for j in range(self._index):
1360            dpmx += self._coeffs[self._index, j] * self.buffer[j]
1361            sum_solver_coeff += self._coeffs[self._index, j]
1362        dpmx += (1 - sum_solver_coeff) * self.buffer[-1]
1363        delta_lamda = lamda_next - lamda
1364        x = (sigma_next / sigma) * x + sigma_next * (delta_lamda) * dpmx
1365        x = x.to(dtype)
1366        self._index += 1
1367        return (x,)
```

## D.2 RECTIFIED FLOW CODE

```
1370 class Scheduler(SchedulerMixin, ConfigMixin):
1371     @register_to_config
1372     def __init__(
1373         self,
1374         num_train_timesteps: int = 1000,
1375         shift: float = 1.0,
1376         use_dynamic_shifting=False,
1377         base_shift: Optional[float] = 0.5,
1378         max_shift: Optional[float] = 1.15,
1379         base_image_seq_len: Optional[int] = 256,
1380         max_image_seq_len: Optional[int] = 4096,
1381     ):
1382         self.num_train_timesteps = num_train_timesteps
1383         self.buffer = []
1384
1385     def set_timesteps(self, sigmas, device: torch.device, *args, **kwargs):
1386         num_inference_steps = len(sigmas)
1387         self._index = 0
1388         self._timedeltas = ...
1389         self._coeffs = ...
1390         self._timesteps = [1.0, ]
1391         for t in range(num_inference_steps - 1):
1392             self._timesteps.append(self._timesteps[-1] - self._timedeltas[t])
1393         self.timesteps = self.timesteps*self.num_train_timesteps
1394         self._timesteps = torch.tensor(self._timesteps)
1395         self.num_inference_steps = num_inference_steps
1396
1397     def step(
1398         self,
1399         v: torch.Tensor,
1400         timestep: int,
1401         x: torch.Tensor,
1402         return_dict: bool = True,
1403     ) -> Union[FlowMatchEulerDiscreteSchedulerOutput, Tuple]:
         if int(timestep) == self.num_train_timesteps:
             self.buffer.clear()
             self._index = 0
         dtype = x.dtype
```

```
dt = self._timedeltas[self._index]
mean = torch.mean(v, [1,], keepdim=True)
v = v - mean
self.buffer.append(v)
v = torch.zeros_like(v)
sum_solver_coeff = 0
for j in range(self._index):
    v += self._coeffs[self._index, j] * self.buffer[j]
    sum_solver_coeff += self._coeffs[self._index, j]
v += (1 - sum_solver_coeff) * self.buffer[-1]
# replace with decayed mean
v = v + mean/(self._index+1)
x = x - v * dt
x = x.to(dtype)
self._index += 1
return (x,)
```

## E    LIMITATIONS

### E.1    MISALIGND RECONSTRUCION LOSS AND PERFORMANCE.

Our proposed methods are specifically designed to minimize integral error within a limited number of steps. However, ablation studies reveal a mismatch between FID performance and Reconstruction error. To address this issue, we plan to enhance our searched solver by incorporating distribution matching supervision, thereby better aligning sampling quality.

### E.2    LARGER CFG INFERENCE.

In the main paper, we demonstrate text-to-image visualization with a small CFG value. However, it is intuitive that utilizing a larger CFG would result in superior image quality. We attribute the inferior performance of large CFGs on our solver to the limitations of current naive solver structures and searching techniques. We hypothesize that incorporating predictor-corrector solver structures would enhance numerical stability and yield better images. Additionally, training with CFGs may also be beneficial.

