# OpenReview forum: "Differentiable Solver Search for fast diffusion sampling"
_ICLR.cc/2025/Conference — Submitted to ICLR 2025_

### Official Review · Reviewer_T3kt · 2024-10-29

**Soundness:** 2
**Presentation:** 1
**Contribution:** 2
**Rating:** 5
**Confidence:** 2

**Summary:**

The paper proposes a method for accelerating reverse diffusion. State-of-the-art models solely rely on the time variable to interpolate and reverse diffuse. The proposed approach builds on the Taylor expansion on top of which the Adams-Bashforth is built around x and not only t in order to improve the search performance. Authors elaborate on the theoretical grounding of their approach and show results on a few benchmarks.

**Strengths:**

The idea of relying on x in addition to t to expand the search space seems very natural.

**Weaknesses:**

The paper's writing is an obstacle for the reader to access the work. The number of typos is too large for me to report them here. There are numerous sudden jumps in the text which miss any logical connectors. Also too many of these to start reporting them.

The analysis in Eq. (7) --> (24) is interesting but it is hard to follow as it is written in a semi-narrative style. It may help to rephrase it as a theorem (state the final result) and the analysis would be the proof of the result.

**Questions:**

What is the computational complexity of the proposed approach and how does it compare to existing methods?

---

> ### Author Response · Authors · 2024-11-15
>
> **Thanks for the valuable feedback**
>
> Thank you for taking the time to provide your valuable feedback. We apologize sincerely for the typos and sudden jumps in the text. We apologize any inconvenience they may have caused. We will thoroughly review every detail and submit a revised version that meets the ICLR standards.
>
> **Q1. Computational complexity compared to other methods.**
>
> **For sampling** When performing sampling over $n$ time steps, our solver caches all pre-sampled predictions, resulting in a memory complexity of  $\mathcal{O}(n)$. The model function evaluation also has a complexity of $\mathcal{O}(n)$ ($\mathcal{O}(2 \times n)$ for CFG enabled). It is important to note that the memory required for caching predictions is negligible compared to that used by model weights and activations. Besides classic methods, we have also included a comparison with the latest Flowturbo published on NeurIPS24.
> |              | Steps | NFE  | NFE-CFG | Cache Pred | Order | search samples   |
> |--------------|-------|------|---------|------------|-------|------------------|
> | Adam2        | n     | n    | 2n      | 2          | 2     | /                |
> | Adam4        | n     | n    | 2n      | 4          | 4     | /                |
> | Henu         | n     | 2n   | 4n      | 2          | 2     | /                |
> | DPM-Solver++ | n     | n    | 2n      | 2          | 2     | /                |
> | UniPC        | n     | n    | 2n      | 3          | 3     | /                |
> | FlowTurbo    | n     | $>$n | $>$2n   | 2          | 2     | 540000(Real)     |
> | our          | n     | n    | 2n      | n          | n     | 50000(Generated) |
>
> **For Searching**  Solver-based algorithms, limited by their searchable parameter sizes, demonstrate significantly lower performance in few-step settings compared to distillation-based algorithms(5/6steps), making direct comparisons inappropriate. Consequently, we selected algorithms that are both acceleratable on ImageNet and comparable in performance, including popular methods such as DPM-Solver++, UniPC(reported in main paper Tab1 and Tab.2), and classic Adams-like linear multi-step methods. Since our experiments primarily utilize SiT, DiT, and FlowDCN that trained on the ImageNet dataset. We also provide fair comparisons by incorporating the latest acceleration method, FlowTurbo. Additionally, we have included results from the Henu method as reported in FlowTurbo.
>
> | SiT-XL-R256 | Steps | NFE-CFG  | Extra-Paramters | FID  | IS    | PR   | Recall |
> |-------------|-------|----------|-----------------|------|-------|------|--------|
> | Heun        | 8     | 16x2     | 0               | 3.68 | /     | /    | /      |
> | Heun        | 11    | 22x2     | 0               | 2.79 | /     | /    | /      |
> | Heun        | 15    | 30x2     | 0               | 2.42 | /     | /    | /      |
> | Adam2 | 16 | 16x2 | 0 | 2.42 | 237 | 0.80 | 0.60 |
> | Adam4 | 16 | 16x2 | 0 | 2.27 | 243 | 0.80 | 0.60 |
> | FlowTurbo   | 6     | (7+3)x2  | 30408704(29M)   | 3.93 | 223.6 | 0.79 | 0.56   |
> | FlowTurbo   | 8     | (8+2)x2  | 30408704(29M)   | 3.63 | /     | /    | /      |
> | FlowTurbo   | 10    | (12+2)x2 | 30408704(29M)   | 2.69 | /     | /    | /      |
> | FlowTurbo   | 15    | (17+3)x2 | 30408704(29M)   | 2.22 | 248    | 0.81    | 0.60      |
> | ours        | 6     | 6x2      | 21              | 3.57 | 214   | 0.77 | 0.58   |
> | ours        | 7     | 7x2      | 28              | 2.78 | 229   | 0.79 | 0.60   |
> | ours        | 8     | 8x2      | 36              | 2.65 | 234   | 0.79 | 0.60   |
> | ours        | 10    | 10x2     | 55              | 2.40 | 238   | 0.79 | 0.60   |
> | ours        | 15    | 15x2     | 110              | 2.24 | 244   | 0.80 | 0.60   |
>
> We can achieve **better or comparable performance** with **much fewer NFE and parameters** compared to FlowTurbo.
>
> **Reference**
>
> [1]. Zhao, Wenliang, et al. "FlowTurbo: Towards Real-time Flow-Based Image Generation with Velocity Refiner." arXiv preprint arXiv:2409.18128 (2024)

---

> > ### Comment · Reviewer_T3kt · 2024-11-27
> >
> > I have read the authors response. I maintain my vote with a low confidence, mainly because of the paper's presentation and writing style which have too many gaps for publication at this point.

---

> > > ### Author Response · Authors · 2024-11-27
> > >
> > > **Presentation Issues**
> > >
> > > We sincerely apologize for the typos in the original submission and any inconvenience they may have caused. We have thoroughly reviewed every detail and submitted a new revised PDF.
> > >
> > > * We thoroughly checked and rectified existing typos, improving the article's readability.
> > > * According to your suggestions, we eliminated most of the redundant formulas and rephrased them as a theorem to maintain the article's clarity.
> > >
> > > We appreciate it greatly that you have offered so many valuable suggestions for our writing. We have resubmitted a revised version of the article to enhance its display quality. And if you have any feedback or suggestions for further improvement, please don't hesitate to contact me directly.

---

> ### Author Response · Authors · 2024-11-15
>
> **Presentation issues**
>
> Thank you for your insightful feedback. We have taken your suggestions and made significant revisions to the article. To maintain clarity, we eliminated most of the redundant formulas and introduced theorems. The proof of theorems is in the appendix.

---

> ### Author Response · Authors · 2024-11-25
>
> As the discussion period ends on November 26, we want to ensure that all your questions have been addressed. Your feedback is invaluable to us, and we would be deeply grateful if you could take a moment to provide a final rating and share your thoughts.

---

### Official Review · Reviewer_j9Yo · 2024-11-01

**Soundness:** 3
**Presentation:** 4
**Contribution:** 3
**Rating:** 6
**Confidence:** 3

**Summary:**

This paper studies diffusion algorithms for generating images. The authors propose a novel differentiable solver search
algorithm to build better diffusion solvers. Specifically, the authors demonstrate that the upper bound of discretization error in reverse-diffusion ODE is related to both timesteps and solver coefficients and define a compact solver search space. Then, a differentiable solver search algorithm can be designed to make better diffusion models. The authors conduct experiments compared with current state-of-the-art methods. They show that the proposed DiT-XL achieves 2.33 FID under ten steps, beating current best methods by a large margin.

**Strengths:**

1. The authors propose a novel differentiable solver search algorithm to build better diffusion solvers. Specifically, the authors demonstrate that the upper bound of discretization error in reverse-diffusion ODE relates to both timesteps and solver coefficients and defines a compact solver search space.

2. The experimental results seem great compared with current state-of-the-art methods.

**Weaknesses:**

FYI: Since I am not working in this area, my reviews may be biased (or even wrong) in a large probability. In general, I found the experimental results to be excellent, and the proposed method seems simple and elegant. I will lean to accept but keep open during the discussion period.

1. The concern of the error bound analysis in Section 4.3: First of all, there are some typos; these $x$ and $\hat{x}$ should be bold. I lost in Equ. (22), should be $||$ be $\| \|_2^2$. The bound provided in Equ. (24) is meaningless to me. It could be helpful to discuss this further. I feel that the authors want to make their method theoretically sound, but it goes in the opposite direction... Even if the authors claim the method is optimal, the algorithm derivation is largely empirical. (Can you justify why the method is optimal? From my understanding, the method should at least match an existing lower bound for the problem.) So, the authors may prefer to keep it as it is.

2. What is $\eta$ in Section 4.3?

3. Section 5 provides algorithms 1 and 2, the proposed differentiable method for solving ODE. This kind of configuration reminds me of some typical extrapolation methods for solving ODE. For example, Richardson's extrapolation for solving ODE forms a kind of table; the method will converge to ODE in a very efficient way. If possible, please discuss this.

**Questions:**

See the weakness.

---

> ### Author Response · Authors · 2024-11-15
>
> **Thanks for valuable feedback**
>
> We appreciate the time you took to share your valuable feedback with us. We offer our sincerest apologies for the typos and any inconvenience they may have caused. We will conduct a thorough review of every detail and submit a revised version that meets the highest standards of the ICLR.
>
> **Q.1 Why do we need to prove that the error bound is related to timesteps and coefficients?**
>
> Our primary objective is to design a compact search space that enables the identification of a solver that achieves near-optimal performance. To accomplish this, we must first establish the constituent components of the search space for the optimal solution. Notably, if the error bound is independent of the number of steps, our search can be limited to the coefficients alone. In fact, it can be proved that the error bound is dictated by the time selection and the coefficients.
>
> **Q.2 What is $\eta$ in Section 4.3?**
>
> $\eta$ is a constant scalar. We will add more explanation of notations in the finial version.
>
>  **Assumption.2** As shown below, the pre-trained velocity model $v_\theta$ is not perfect and the error between ${v}_\theta$ and ideal velocity field $\hat{v}$ is bounded, where $\eta$ is a constant scalar.
>
> $||\hat{v}-v_\theta || \leq \eta \ll ||\hat{v}|| $
>
> **Q.3 Richardson's extrapolation for solving ODE**
>
> Yes, the Adams-like linear multi-step method employs Lagrange interpolation to determine its coefficients, which makes it feasible to substitute Lagrange interpolation with alternative interpolation (or extrapolation) techniques[1], such as Richardson's method. Nevertheless, Richardson functions also solely rely on the variable $t$, without considering $x$.
>
> **Reference**
>
> [1] Fekete, Imre, and Lajos Lóczi. "Linear multistep methods and global Richardson extrapolation." Applied Mathematics Letters 133 (2022): 108267.

---

> > ### Comment · Reviewer_j9Yo · 2024-11-25
> > **I will keep my score.**
> >
> > I have read other reviewers' comments and the authors' responses. The proposed algorithm is a nice modification of the conventional algorithm, such as the numerical ODE solvers. This new algorithm is simple and works well. Therefore, I will keep my score unchanged and lean toward acceptance.

---

### Official Review · Reviewer_D9P2 · 2024-11-04

**Soundness:** 2
**Presentation:** 2
**Contribution:** 3
**Rating:** 8
**Confidence:** 2

**Summary:**

This paper addresses the inefficiencies in diffusion models for image generation, which require numerous denoising steps during inference. The authors present several key contributions:

1. The authors demonstrate that the choice of interpolation function in the reverse-diffusion ODE can be reduced to mere coefficients, which simplifies the error minimization process related to discretization.

2. The authors propose a novel algorithm that identifies optimal solver parameters within a compact search space defined by timesteps and solver coefficients, enhancing the performance of pre-trained diffusion models.

3. Utilizing their algorithm, they achieve state-of-the-art (relative to a selection of methods) results on ImageNet from 5 to 10 sampling steps.

**Strengths:**

# Content

1. The paper critically revisits Adams-like multistep methods and highlights their limitations specifically in the context of diffusion models.

2. The derivation of error bounds and the use of Cauchy-Schwarz inequalities to establish relationships between error, solver coefficients, and timestep choices demonstrate a rigorous mathematical approach.

3. By proposing a universal interpolation function $\mathcal{P}$ without an explicit form and focusing on coefficients rather than fixed interpolation methods, the paper opens new avenues for flexibility in solver design. This could lead to more adaptable and potentially more accurate methods in sampling the reverse diffusion process.

4. The introduction of a differentiable solver search algorithm provides a novel way to optimize timesteps and coefficients. This approach could leverage pre-trained models, possibly leading to improved performance in practical applications.

5. The paper's focus on error bounds related to pre-trained velocity models is valuable, as it acknowledges the imperfections in real-world applications and provides a framework for quantifying these errors.

**Weaknesses:**

# Presentation (Minor)

I marked the Presentation as poor. The reason for this is that, to my liking, the equations are not properly embedded into the text and there are too many prominent typos.

Please improve your usage of punctuation in and surrounding equations. Furthermore, 29 enumerated equations in the main paper, of which many are not referenced, can be considered excessive. Detailed derivations could be moved to the appendix, shifting the focus to the core functions of your method and leaving more space for figures 4 & 5 (e.g. allowing for larger text within the figures), and algorithms 1 & 2. This could drastically improve the presentation of your work.

To further improve the presentation of your work, please also check for typos, like in the title of section 3, Eular vs. Euler, etc..

# Content (Major)

1. The emphasis on optimizing solver coefficients based on small data (50K in the experiment section) raises concerns about overfitting. While the expectation of coefficients is meant to enhance generalization, the process must be carefully managed to ensure robust performance across varied datasets.

2. The Paper does not feature any other metrics than FID.

3. While the paper suggests state-of-the-art performance, its experiments and comparisons appear selective. It is important to compare it to other methods that could potentially outperform your method as well. Otherwise, the reader has no perspective regarding the limitations of your approach.

4. The Paper also does not discuss limitations w.r.t. how well the solver algorithm scales to smaller or larger amounts of samples. Furthermore, all evaluation was stopped at 5 solver steps.

**Questions:**

In general, I am willing to raise my score if my questions and concerns are addressed with compelling evidence.

Concerning the aforementioned weaknesses, I pose the following questions:

1. The paper features FID as its only metric. Can you incorporate more metrics, such as e.g. Improved Precision & Recall, as well as Inception-Score?

2. How long does it take for Algorithm 2 to complete in theory? O-Noation w.r.t. network evaluations, samples and solver steps should be featured in your paper.

3. You used 50K samples for Algorithm 2 in your experiments section, can you add an ablation study for the cardinality of the samples used to solve your coefficient search? (e.g. 10K, 50K, 100K & 1M samples)

4. You stopped your evaluation at 5 Steps, how much do scores deteriorate for 1 to 4 steps, can you add an additional ablation study for less than 5 solver steps?

5. While your evaluation in Tables 1 & 2 suggests your method outperforms competing methods, how does your work compare to Distillation Methods, such as Consistency-Distillation Training, which yields methods that require less than 5 solver steps? Such comparisons should be featured to put the performance of your method into perspective relative to the state-of-the-art for efficient solving techniques of the reverse process.

6. How do you explain the 10-step solver outperforming 50 Euler steps in Figure 5 (c), what scores would your method reach for 50 steps? I kindly ask you to evaluate more than one metric (see 1.).

7. How well does your method work across different variance schedules? Can variance schedules be identified, where your method works better or worse? Does your method perform better on diffusion processes where the forward process is driftless (e.g. VE) or forward processes that do not omit the drift function (e.g. VP)?

8. Can you add an evaluation of your text-to-image experiments that is based on metrics rather than visual impressions?

Overall I kindly ask you to rework your paper's presentation and add a more rigorous evaluation with more metrics than FID, measuring the diversity- and fidelity of samples.

---

> ### Author Response · Authors · 2024-11-15
>
> **Thanks for the valuable feedback**
>
> Thank you for taking the time to provide your valuable feedback. We apologize sincerely for the typos and any inconvenience they may have caused. We will thoroughly review every detail and submit a revised version that meets the ICLR standards.
>
> **Q1. More Metrics of Searched Solver**
>
> We adhere to the evaluation guidelines provided by DM-nonuniform, reporting only the FID as the standard metric in the current main paper.
>
> To clarify, we do not report selective results; **we will provide sFID, IS, PR, and Recall metrics for SiT-XL(R256), FlowDCN-XL/2(R256), and FlowDCN-B/2(R256) in a new revision pdf (cause we can not directly submit figs on openreview)**. Our solver searched on FlowDCN-B/2, consistently outperforms the handcrafted solvers across FID, sFID, IS, and Recall metrics.
>
> **Q2. Computational complexity compared to other methods.**
>
> **For sampling** When performing sampling over $n$ time steps, our solver caches all pre-sampled predictions, resulting in a memory complexity of  $\mathcal{O}(n)$. The model function evaluation also has a complexity of $\mathcal{O}(n)$ ($\mathcal{O}(2 \times n)$ for CFG enabled). It is important to note that the memory required for caching predictions is negligible compared to that used by model weights and activations. Besides classic methods, we have also included a comparison with the latest Flowturbo published on NeurIPS24.
> |              | Steps | NFE  | NFE-CFG | Cache Pred | Order | search samples   |
> |--------------|-------|------|---------|------------|-------|------------------|
> | Adam2        | n     | n    | 2n      | 2          | 2     | /                |
> | Adam4        | n     | n    | 2n      | 4          | 4     | /                |
> | Heun         | n     | 2n   | 4n      | 2          | 2     | /                |
> | DPM-Solver++ | n     | n    | 2n      | 2          | 2     | /                |
> | UniPC        | n     | n    | 2n      | 3          | 3     | /                |
> | FlowTurbo    | n     | $>$n | $>$2n   | 2          | 2     | 540000(Real)     |
> | our          | n     | n    | 2n      | n          | n     | 50000(Generated) |
>
> **For Searching**  Solver-based algorithms, limited by their searchable parameter sizes, demonstrate significantly lower performance in few-step settings compared to distillation-based algorithms(5/6steps), making direct comparisons inappropriate. Consequently, we selected algorithms that are both acceleratable on ImageNet and comparable in performance, including popular methods such as DPM-Solver++, UniPC(reported in main paper Tab1 and Tab.2), and classic Adams-like linear multi-step methods. Since our experiments primarily utilize SiT, DiT, and FlowDCN that trained on the ImageNet dataset. We also provide fair comparisons by incorporating the latest acceleration method, FlowTurbo. Additionally, we have included results from the Heun method as reported in FlowTurbo.
>
> We can achieve **better or comparable performance** with **much fewer NFE and parameters** compared to FlowTurbo.
>
> | SiT-XL-R256 | Steps | NFE-CFG  | Extra-Paramters | FID  | IS    | PR   | Recall |
> |-------------|-------|----------|-----------------|------|-------|------|--------|
> | Heun        | 8     | 16x2     | 0               | 3.68 | /     | /    | /      |
> | Heun        | 11    | 22x2     | 0               | 2.79 | /     | /    | /      |
> | Heun        | 15    | 30x2     | 0               | 2.42 | /     | /    | /      |
> | Adam2 | 16 | 16x2 | 0 | 2.42 | 237 | 0.80 | 0.60 |
> | Adam4 | 16 | 16x2 | 0 | 2.27 | 243 | 0.80 | 0.60 |
> | FlowTurbo   | 6     | (7+3)x2  | 30408704(29M)   | 3.93 | 223.6 | 0.79 | 0.56   |
> | FlowTurbo   | 8     | (8+2)x2  | 30408704(29M)   | 3.63 | /     | /    | /      |
> | FlowTurbo   | 10    | (12+2)x2 | 30408704(29M)   | 2.69 | /     | /    | /      |
> | FlowTurbo   | 15    | (17+3)x2 | 30408704(29M)   | 2.22 | 248    | 0.81    | 0.60      |
> | ours        | 6     | 6x2      | 21              | 3.57 | 214   | 0.77 | 0.58   |
> | ours        | 7     | 7x2      | 28              | 2.78 | 229   | 0.79 | 0.60   |
> | ours        | 8     | 8x2      | 36              | 2.65 | 234   | 0.79 | 0.60   |
> | ours        | 10    | 10x2     | 55              | 2.40 | 238   | 0.79 | 0.60   |
> | ours        | 15    | 15x2     | 110              | 2.24 | 244   | 0.80 | 0.60   |
>
> **Reference**
>
> [1]. Zhao, Wenliang, et al. "FlowTurbo: Towards Real-time Flow-Based Image Generation with Velocity Refiner." arXiv preprint arXiv:2409.18128 (2024)
>
> [2]. Xue, Shuchen, et al. "DM-nonuniform: Accelerating Diffusion Sampling with Optimized Time Steps." Proceedings of the IEEE/CVF Conference on Computer Vision and Pattern Recognition. 2024.

---

> ### Author Response · Authors · 2024-11-15
>
> **Q3. Ablation on Search Samples**
> We ablate the number of search samples on the 10-step and 8-step solver settings. \textit{Samples} means the total training samples the searched solver has seen.   \textit{Unique Samples} means the total distinct samples the searched solver has seen.  Our searched solver converges fast and gets saturated near 30000 samples.
>
> | iters(10-step-solver) | samples | unique samples | FID  | IS  | PR   | Recall |
> |-----------------------|---------|----------------|------|-----|------|--------|
> | 313                   | 10000   | 10000          | 2.54 | 239 | 0.79 | 0.59   |
> | 626                   | 20000   | 10000          | 2.38 | 239 | 0.79 | 0.60   |
> | 939                   | 30000   | 10000          | 2.49 | 240 | 0.79 | 0.59   |
> | 1252                  | 40000   | 10000          | 2.29 | 239 | 0.80 | 0.60   |
> | 1565                  | 50000   | 10000          | 2.41 | 240 | 0.80 | 0.59   |
> | 626                   | 20000   | 20000          | 2.47 | 237 | 0.78 | 0.60   |
> | 939                   | 30000   | 30000          | 2.40 | 238 | 0.79 | 0.60   |
> | 1252                  | 40000   | 40000          | 2.48 | 237 | 0.80 | 0.59   |
> | 1565                  | 50000   | 50000          | 2.41 | 239 | 0.80 | 0.59   |
>
> | iters(8-step-solver) | samples | unique samples | FID  | IS  | PR   | Recall |
> |----------------------|---------|----------------|------|-----|------|--------|
> | 313                  | 10000   | 10000          | 2.99 | 228 | 0.78 | 0.59   |
> | 626                  | 20000   | 10000          | 2.78 | 229 | 0.79 | 0.60   |
> | 939                  | 30000   | 10000          | 2.72 | 235 | 0.79 | 0.60   |
> | 1252                 | 40000   | 10000          | 2.67 | 228 | 0.79 | 0.60   |
> | 1565                 | 50000   | 10000          | 2.69 | 235 | 0.79 | 0.59   |
> | 626                  | 20000   | 20000          | 2.70 | 231 | 0.79 | 0.59   |
> | 939                  | 30000   | 30000          | 2.82 | 232 | 0.79 | 0.59   |
> | 1252                 | 40000   | 40000          | 2.79 | 231 | 0.79 | 0.60   |
> | 1565                 | 50000   | 50000          | 2.65 | 234 | 0.79 | 0.60   |
>
>
> **Q4. Stopped evaluation at 5 Steps.**
>
> Since DM-nonuniform introduced the most effective online optimization solver before our search-based approach, we leveraged their results for comparison on DDPM models. We followed the evaluation pipeline established by DM-nonuniform to report performance within 5 and 10 optimization steps. In general, solver-based methods tend to exhibit inferior results under extremely limited numbers of function evaluations (NFE), such as 5 or 6 steps.
>
> **Q5. comparison with distillation methods**
>
> We provide a comparison with FlowTurbo in Q.2
>
> Under the given NFE (Number of Function Evaluations) condition, Adams-like linear multistep methods are the strongest manually designed solvers, with performance far superior to Heun and RK4, and relevant test results can be found in Q2. So we used the linear multi-step method as a comparison object.
>
> As the solving difficulty increases and the number of searchable parameters decreases (e.g., only 10 searchable parameters for 4 steps and 6 searchable parameters for 3 steps), the performance of solver-based methods falls significantly behind that of distillation methods when limited to fewer than 5 steps. Notably, it is unlikely for solver-based methods to achieve performance comparable to or exceeding that of distillation methods, such as CM, given that their number of learnable parameters is tens of thousands of times larger than our searchable parameters.
>
> Furthermore, integrating denoiser distillation with solver search holds significant promise for achieving even greater performance enhancements.
>
> **Q6. 10-step solver outperforming 50 Euler steps.**
>
> Linear multistep-based high-order solvers can significantly boost performance in simulations with a limited number of time steps. By leveraging the reference trajectory from the Euler solver with 100 steps, it is possible to outperform the Euler solver with 50 steps. As illustrated in all metrics, our solver enables SiT-XL/2-R256 and FlowDCN-XL/2-R256 to achieve better Recall scores than the Euler solver with 50 steps. Notably, FlowDCN-XL/2-R512 with our solver surpasses its Euler counterpart in terms of sFID, Precision, and Recall, demonstrating its exceptional performance.

---

> ### Author Response · Authors · 2024-11-15
>
> **Q7.1. Solver Across different variance schedules**
>
> Since our solvers are searched on a specific noise scheduler and its corresponding pre-trained models, applying the searched coefficients and timesteps to other noise schedulers yields meaningless results. We have tried applied searched solver on SiT(Rectified flow) and DiT(DDPM with $\beta_{min}=0.1, \beta_{max}=20$) to SD1.5(DDPM with $\beta_{min}=0.085, \beta_{max}=12$), but the results were inconclusive. Notably, despite sharing the DDPM name, DiT and SD1.5 employ distinct $\beta_{min}, \beta_{max}$ values, thereby featuring different noise schedulers. A more in-depth discussion of these experiments can be found in Section(Extend to DDPM/VP).
>
> **Q7.2. Solver for different variance schedules**
>
> Since every discrete Denoising Diffusion Probabilistic Model (DDPM) has a corresponding continuous Variance Preserving (VP) scheduler, we can transform the discrete DDPM into a continuous VP, thereby successfully finding a better solver compared to traditional DPM-Solvers.
>
> To put it simply, under the empowerment of our high-order solver, the performance of DDPM and Rectified flow does not differ significantly (8, 9, 10 steps), which contradicts the common belief that Rectified flow is stronger at limited sampling steps.

---

> ### Author Response · Authors · 2024-11-17
> **Supplimentary experiments for Q.4**
>
> **Q4. Stopped evaluation at 5 Steps.**
>
> Since DM-nonuniform introduced the most effective online optimization solver before our search-based approach, we leveraged their results for comparison on DDPM models. We followed the evaluation pipeline established by DM-nonuniform to report performance within 5 and 10 optimization steps. In general, solver-based methods tend to exhibit inferior results under extremely limited numbers of function evaluations (NFE), such as 5 or 6 steps.
>
> We provide the experiments below 5 steps. Note that when reduced to a single step, our algorithm essentially has no parameters and will have exactly the same performance of the Euler solver with 1 step.
> |       | Steps | NFE-CFG | FID   | IS    | PR   | Recall |
> |-------|-------|---------|-------|-------|------|--------|
> | Euler | 1     | 1x2     | 300   | 2.32  | /    | /      |
> | Euler | 50    | 50x2    | 2.23  | 244   | 0.80 | 0.59   |
> | Adam2 | 3     | 3x2     | 41.2  | 68.6  | 0.44 | 0.46   |
> | Adam2 | 4     | 4x2     | 15.25 | 133.6 | 0.65 | 0.50   |
> | Adam2 | 5     | 5x2     | 8.96  | 170   | 0.73 | 0.53   |
> | Adam2 | 6     | 6x2     | 6.35  | 191   | 0.76 | 0.55   |
> | Adam2 | 15    | 15x2    | 2.49  | 236   | 0.79 | 0.59   |
> | Adam4 | 15    | 15x2    | 2.33  | 242   | 0.80 | 0.59   |
> | ours  | 1     | 1x2     | 300   | 2.32  | /    | /      |
> | ours  | 3     | 3x2     | 39.3  | 68.6  | 0.46 | 0.52   |
> | ours  | 4     | 4x2     | 13.9  | 135   | 0.65 | 0.55   |
> | ours  | 5     | 5x2     | 4.52  | 194   | 0.75 | 0.58   |
> | ours  | 6     | 6x2     | 3.57  | 214   | 0.77 | 0.58   |
> | ours  | 15    | 15x2    | 2.24  | 244   | 0.80 | 0.60   |
>
>
> So, theoretically speaking, our algorithm will converge to the same result as the Euler method and Adam-like methods in 1 step and huge sampling steps(eg. 500 steps or 1000 steps), and will significantly outperform these algorithms at intermediate step numbers.

---

> ### Author Response · Authors · 2024-11-20
>
> **Q.8 Text to Image Metrics Result**
>
> We take PixArt-XL-2-256x256[1] as the text-to-image model. We follow the evaluation pipeline of ADM and take COCO17-Val as the reference batch. We generate 5k images using DPM-Solver++, UniPC, and our solver(searched on DiT-XL/2-R256).
>
> Our method consistently achieves better FID metrics results.
>
> |       | Steps | FID  | sFID  | IS    | PR   | Recall |
> |-------|-------|------|-------|-------|------|--------|
> | DPM++ | 5     | 60.0 | 209   | 25.59 | 0.36 | 0.20   |
> | DPM++ | 8     | 38.4 | 116.9 | 33.0  | 0.50 | 0.36   |
> | DPM++ | 10    | 35.6 | 114.7 | 33.7  | 0.53 | 0.37   |
> | UniPC | 5     | 57.9 | 206.4 | 25.88 | 0.38 | 0.20   |
> | UniPC | 8     | 37.6 | 115.3 | 33.3  | 0.51 | 0.36   |
> | UniPC | 10    | 35.3 | 113.3 | 33.6  | 0.54 | 0.36   |
> | Ours  | 5     | 46.4 | 204   | 28.0  | 0.46 | 0.23   |
> | Ours  | 8     | 33.6 | 115.2 | 32.6  | 0.54 | 0.39   |
> | Ours  | 10    | 33.4 | 114.7 | 32.5  | 0.55 | 0.39   |
>
> [1]. Chen, Junsong, et al. "Pixart-$\alpha $: Fast training of diffusion transformer for photorealistic text-to-image synthesis." arXiv preprint arXiv:2310.00426 (2023).

---

> ### Author Response · Authors · 2024-11-24
>
> As the discussion period ends on November 26, we want to ensure that all your questions have been thoroughly addressed.  Your feedback is instrumental to us, and we would be grateful if you could spare a moment to provide a final rating and share your thoughts. Your input will greatly inform our future improvements.

---

> > ### Comment · Reviewer_D9P2 · 2024-11-24
> >
> > I acknowledge that the authors have gone to great lengths to address my concerns. Though not every point was addressed with additional experiments, all of my points have been addressed to the best of the authors' abilities.
> >
> > Therefore I have reconsidered my initial assessment and raised my **rating to an (8)** (accept).
> >
> > However, I also lowered my **confidence to a (2)**, as I believe that extending the experiments of the method to other Variance Schedules, such as the simple VE schedule or a variance schedule with hyperparameters such as in EDM would have provided more confidence to defend my assessment.
> >
> > This is my final assessment of the paper.

---

### Author Response · Authors · 2024-11-15

**Presentation issues**

Thank all three reviewers for taking the time to provide valuable comments. We apologize sincerely for the typos and any inconvenience they may have caused. We have thoroughly reviewed every detail and submitted a new revised version.

* We thoroughly checked and rectified existing typos, improving the article's readability.
* We eliminated most of the redundant formulas and introduced theorems to maintain the article's clarity.

We have resubmitted a revised version of the article, with the aim of enhancing its display quality. If the reviewers have any constructive feedback or suggestions for further improvement, please don't hesitate to reach out to me directly.

---

### Author Response · Authors · 2024-11-24

As the discussion period ends on November 26, we are eager to ensure that all the questions have been thoroughly resolved. We hope that our responses have adequately addressed your concerns. Your feedback is invaluable to us, and we would greatly appreciate it if you could take a moment to provide a final rating and feedback.

---

### Meta-Review · Area_Chair_oguQ · 2024-12-19

**Metareview:**

This work is on the topic of fast sampling of diffusion models. The authors parametrize a class of ODE solvers and then optimize these parameters for fast sampling. This additional optimization step is the major difference to most existing training-free sampling algorithms for diffusion models. The paper also presents some theoretical results to analyze the performance of the proposed algorithm. The reviewers raises some questions on the theoretical and experimental results, as well as the presentation of the paper. The theoretical results (Theorem 4,4, 4.5) appear to be straightforward from textbook. Similar results have also been established in the diffusion model literatures; see e.g. [1][2]. In addition, even though the authors argue that the coefficients in (14) should depend on x for better performance, the final algorithm developed in this paper reduces to standard ODE solver as shown in (15). That being said, the only difference between this work and existing ODE methods for diffusion models is that the solver coefficients and time discretization points are now trainable. Moreover, the experiment is not comprehensive enough with many baselines and metrics missing. It doesn’t show the advantages of the proposed algorithms to solve ODE. Finally, since the proposed algorithm is no longer training free, distillation type algorithms should also be taken into account as baselines.
[1] DPM-SOLVER++: FAST SOLVER FOR GUIDED SAMPLING OF DIFFUSION PROBABILISTIC MODELS
[2] Improved Order Analysis and Design of Exponential Integrator for Diffusion Models Sampling

**Additional Comments On Reviewer Discussion:**

The reviewers raises some questions on the theoretical and experimental results, as well as the presentation of the paper. The authors reply by modifying the paper, adding experiments in the paper, and adding clarifications in the response. Overall, the reviewers are not excited about the paper  and several of them choose to keep their original evaluation.

---

### Decision · Program_Chairs · 2025-01-22

Reject